# A resource for the *Drosophila* antennal lobe provided by the connectome of glomerulus VA1v

Jane Anne Horne[1], Carlie Langille[1], Sari McLin[1], Meagan Wiederman[1], Zhiyuan Lu[1,2], C Shan Xu[2], Stephen M Plaza[2], Louis K Scheffer[2], Harald F Hess[2], Ian A Meinertzhagen[1,2]*

[1]Department of Psychology and Neuroscience, Life Sciences Centre, Dalhousie University, Halifax, Canada; [2]Janelia Research Campus, Howard Hughes Medical Institute, Virginia, United States

**Abstract** Using FIB-SEM we report the entire synaptic connectome of glomerulus VA1v of the right antennal lobe in *Drosophila melanogaster*. Within the glomerulus we densely reconstructed all neurons, including hitherto elusive local interneurons. The *fruitless*-positive, sexually dimorphic VA1v included >11,140 presynaptic sites with ~38,050 postsynaptic dendrites. These connected input olfactory receptor neurons (ORNs, 51 ipsilateral, 56 contralateral), output projection neurons (18 PNs), and local interneurons (56 of >150 previously reported LNs). ORNs are predominantly presynaptic and PNs predominantly postsynaptic; newly reported LN circuits are largely an equal mixture and confer extensive synaptic reciprocity, except the newly reported LN2V with input from ORNs and outputs mostly to monoglomerular PNs, however. PNs were more numerous than previously reported from genetic screens, suggesting that the latter failed to reach saturation. We report a matrix of 192 bodies each having $\geq$50 connections; these form 88% of the glomerulus' pre/postsynaptic sites.
DOI: https://doi.org/10.7554/eLife.37550.001

*For correspondence:
I.A.Meinertzhagen@Dal.Ca

**Competing interests:** The authors declare that no competing interests exist.

## Introduction

A striking convergence in advanced brains has endowed those structures of the brains in insects and mammals that analyse olfactory information with close similarities (*Hildebrand and Shepherd, 1997*). In both, segregated modules of neuropile called glomeruli each receive input from the axons of olfactory receptor neurons (ORNs), each of which in turn expresses predominantly a single olfactory receptor gene (*Gao et al., 2000*; *Vosshall et al., 2000*). ORNs in the adult fruit fly *Drosophila melanogaster* belong to one of four classes of sensilla that form a regular pattern on the third segment of the fly's antenna and maxillary palp (*Vosshall and Stocker, 2007*). Compared with the ~$5\times10^6$ ORNs (*Kawagishi et al., 2014*) that innervate the ~1800 glomeruli in the mouse olfactory bulb (*Royet et al., 1988*), the different regions of the olfactory system in *Drosophila* signal with fewer cell types using fewer odorant receptor molecules. Thus, only 1300 ORNs that express 62 odorant receptor proteins (*Vosshall and Stocker, 2007*) innervate a mere ~50 modular glomeruli (*Grabe et al., 2015*), the first relay of the insect olfactory system (*Stocker, 1994*; *Laissue et al., 1999*; *Gao et al., 2000*; *Vosshall et al., 2000*; *Benton et al., 2009*; *Grabe et al., 2015*).

Each of the ~50 glomeruli has been individually identified (*Laissue et al., 1999*; *Benton et al., 2009*; *Silbering et al., 2011*) and mapped, both in vitro, after dissection (*Stocker et al., 1983*; *Laissue et al., 1999*; *Couto et al., 2005*; *Endo et al., 2007*; *Silbering et al., 2011*) and in vivo (*Grabe et al., 2015*). Output from the glomeruli is made by the antennal lobe projection neurons (PNs), some of which relay olfactory information to higher-order olfactory centres, the mushroom

body and lateral horn (*Wong et al., 2002*; *Marin et al., 2002*; *Yasuyama et al., 2002*; *Yasuyama et al., 2003*) via three main antennal lobe tracts (ALTs), medial, mediolateral and lateral (mALT, mlALT and lALT).

The cellular composition of the antennal lobe has also been extensively reported, both from early back-fill studies (*Stocker et al., 1990*) and more recent genetic reporter lines (e.g. *Tanaka et al., 2012*), and the numbers, types and patterns of innervation these receive from ORNs has likewise been identified from such lines (e.g. *Couto et al., 2005*). Among the PNs, *Tanaka et al. (2012)* have identified eleven classes, four in the medial, three in the mediolateral, and four in the lateral tracts. Amongst these, mPN1s project out of the antennal lobe along the mALT to the mushroom body and lateral horn (*Tanaka et al., 2012*). Relative to these, ORNs have been morphologically less well characterized but are recently catalogued for glomerulus DM6 (*Tobin et al., 2017*).

Little studied hitherto are the local interneurons (LNs), a major focus of our study. Tanaka has identified LNs of six types with cell bodies in different locations, which arborize in either single or multiple glomeruli and in the antennal lobe of either one or both sides (*Table 1*). Categorized by the glomeruli they innervate, *Chou et al. (2010)* identify at least ~100 different LNs for each side. Additional cell types identified include a single 5-HT neuron and several transverse (tALT) neurons (*Tanaka et al., 2012*).

Despite detailed knowledge of their cellular composition, the synaptic networks of the fly's antennal lobe have been incompletely documented at the requisite level from electron microscopy (EM). The reasons for this are obvious: the difficulty of tracing the tiny neurites of *Drosophila* neurons, the labour required to do so, and the practical problems of targeting specific glomeruli for EM. Fluorescent pre- and postsynaptic markers provide proxies for claimed synaptic contacts (*Mosca and Luo, 2014*), but lack independent validation from EM. Two recent studies (*Rybak et al., 2016*; *Tobin et al., 2017*) report the synaptic networks of identified glomeruli. The first, a study of three glomeruli (DL5, DM2, and VA7: *Laissue et al., 1999*), has provided a significant map of synaptic connections and their numerical proportions, while the second, on glomerulus DM6 (*Tobin et al., 2017*), established many features of the PN. Neither study gave a comprehensive account of the varied synaptic inputs from and between LNs, however. Including the synaptic network of all LNs and PNs to generate an actual map of the complete synaptic network, or connectome (*Lichtman and Sanes, 2008*), for each glomerulus is thus a final step towards adopting functional connectomic approaches (*Venken et al., 2011*; *Meinertzhagen and Lee, 2012*) to the analysis of antennal lobe function.

## Results

We report the connectome of a single glomerulus VA1v of the right side of a female *Drosophila* antennal lobe (*Figure 1*), using dense reconstruction of an isotropic 8 nm voxel image stack (*Figure 2*), see Materials and methods, obtained by focused ion beam (FIB) milling scanning electron microscopy, FIB-SEM (*Xu et al., 2017*). VA1v is a sexually dimorphic *fruitless*-positive glomerulus responsive to fly odour (*Sakurai et al., 2013*), that signals the sex pheromones cis-vaccenyl acetate and methyl laurate (*Kurtovic et al., 2007*; *Dweck et al., 2013*), and is significantly larger in male flies than in females (*Kondoh et al., 2003*; *Stockinger et al., 2005*).

### Synapses

Each synapse comprised a T-bar ribbon at the presynaptic site opposite membrane densities at the postsynaptic dendrites (*Figure 3A*). Most commonly these numbered three per synapse, with an average of 3.4, most synapses thus forming triads or tetrads (*Figure 2—source data 1*), like those of the three glomeruli reported by *Rybak et al. (2016)*. The range was from 1:1 to 1:9. The numbers of postsynaptic dendrites differed slightly for each presynaptic cell, ipsilateral ORN cells having 3.63 dendrites per synapse, LN1 and LN2L 3.17 and 3.06 respectively, and mPN1 had 3.54 postsynaptic dendrites (*Figure 2—source data 1*).

The size and structural features of each synapse resembled those found between neurons in other *Drosophila* neuropiles, such as the optic lobe lamina and medulla (*Meinertzhagen and O'Neil, 1991*; *Takemura et al., 2008*). At each presynaptic site a T-bar was found with a clear pedestal; the electron density of the ribbon platform surmounting it was weak after the preparation methods adopted for FIB-SEM. Thus unlike transmission EM, TEM (*Rybak et al., 2016*), ribbons often lacked canonical T-bar profiles when seen in FIB-SEM images. This chiefly resulted because in our FIB-SEM

**Table 1.** Nomenclatures for antennal lobe neurons

| [*]Tanaka et al. (2012) | Gal four enhancer-trap strain | Cell count | Va1v cells, this study |
|---|---|---|---|
| LN1 | NP1227 | 18 | 17 |
| LN2L | NP2426 | 40 | 24 |
| LN2V[‡] | NP2427 | 3 | 1 |
| LN3 | NP1326 | 4.5 | 1 |
| LN4 | NP842 | 3 | 1 |
| LN6 | NP1587 | 1 | 2 |
| mPN1[†] | | 59 | 3 |
| mlPN1 | NP5288 | 9.5 | 2 |
| mlPN2 | | 23 | 3 |
| mlPN3 | | 3 | 2 |
| t3PN1 | | 2 | |
| t4PN1 | | 5.5 | |
| lPN2 | | 9.3 | 3 |
| lPN4 | | 1 | 3 |
| AST1 | | 1 | |
| AST2 | | 1 | |
| AST3 | | 1 | 1 |
| MBDL1 | | 1 | |
| VUMa5 | | 1 | |
| 5HT | | 1 | 1[§] |
| | | | |
| Cells not reconciled with those reported by *Tanaka et al. (2012)* | | | |
| LNV | | | 3 |
| LN_LVExit | | | 3 |
| LN_commissure | | | 4 |
| Total | | | 10 |
| Ipsilateral ORNs | | | 51 |
| contralateral ORNs | | | 56 |

[*]Column lists all those cells reported by *Tanaka et al. (2012)* to enter Va1v

**[†]Tanaka labelled only non-specifically for several cell classes including those in Va1v

***[‡]Tanaka did not find in Va1v at all

****[§]Not included in matrix

DOI: https://doi.org/10.7554/eLife.37550.002

images the grey level of the synapses was about twice that of the membranes. We therefore adjusted the electron density of images to increase membrane contrast, because this proved advantageous to enhance membrane continuity more reliably during later proof-reading steps (see Materials and methods), but rendered the platform of the T-bar ribbon often less distinct in our FIB-SEM images than when seen in TEM.

Occasional synapses were bidirectional, with a T-bar ribbon on either side of the synaptic cleft. Synapse sizes also varied more widely than for tetrad synapses in the lamina (*Fröhlich, 1985*; *Meinertzhagen and O'Neil, 1991*). Occasional autapses were seen amongst PN axons, at which the neuron was presynaptic to itself. These typically numbered about 1% of all the PN axon's synapses, about the same relative number as reported for medulla neurons (*Takemura et al., 2015*). Synaptic vesicles were not well resolved after FIB-SEM imaging at 8 nm. In addition to synapses with what appear to be small, clear vesicles, dark vesicles having a dense core of between 50 and 100 nm in

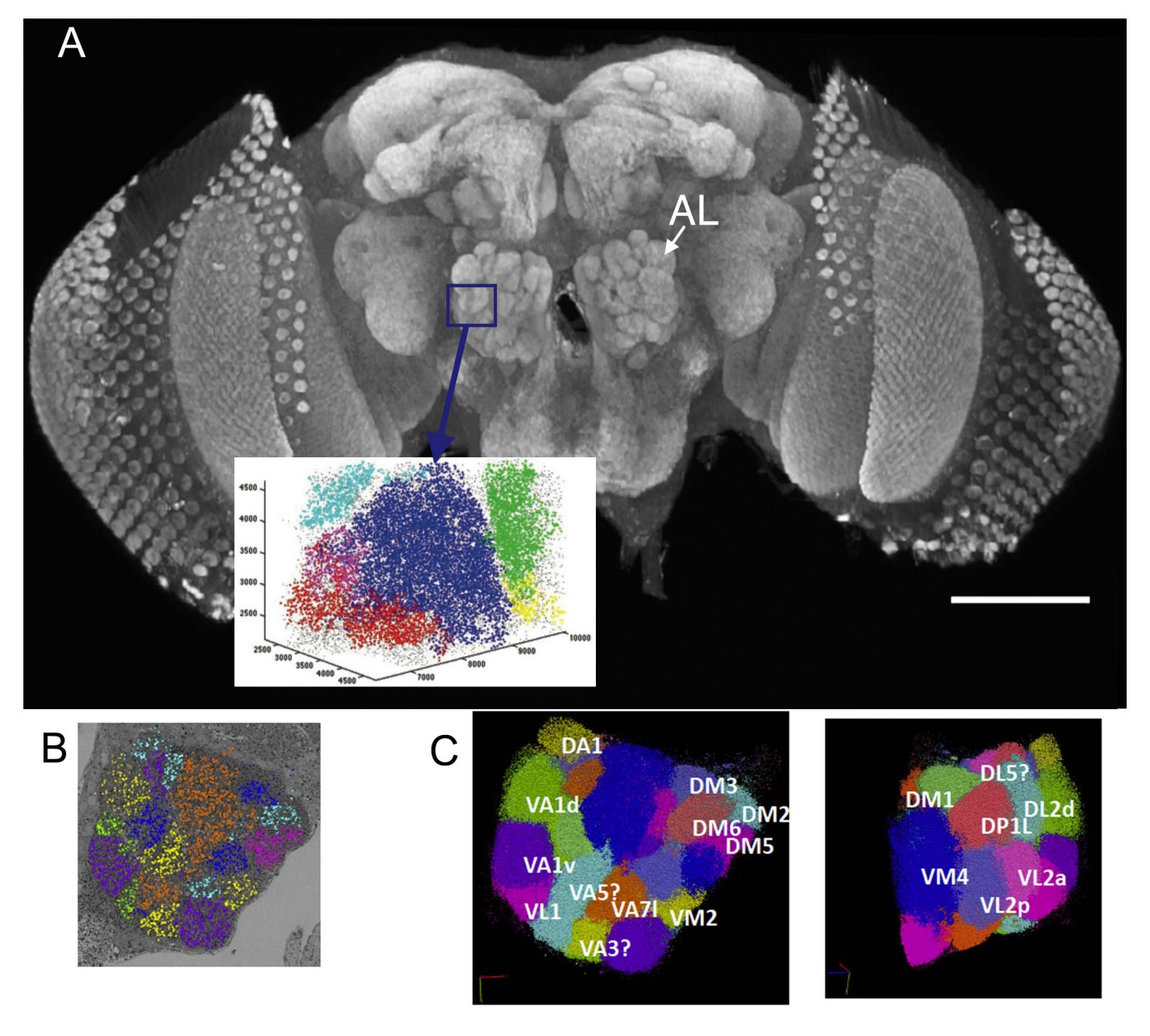

**Figure 1.** (A) The right antennal lobe of a female *Drosophila melanogater* wholemounted and immunolabelled with nc82 to detect Bruchpilot at synaptic sites, revealing the neuropiles of the brain. These included the glomeruli of the antennal lobe (AL). Va1v of the right antennal lobe (blue) is enclosed by a box in A, identified in synapse cloud images. Scale bar 100 µm. (B) Image of a single plane with synapses identified as clouds by synapse detection on a corresponding FIB-SEM image stack, enclosing most of the right antennal lobe within which different glomeruli are identified and colour coded. Glomerular borders are visible in this single image from local rarefactions in the density of recognized synapse puncta. (C) A cloud of synaptic puncta in two image planes parallel and corresponding to the one in (B), with different glomeruli – including VA1v (left panel) – identified.
DOI: https://doi.org/10.7554/eLife.37550.003

diameter appeared in the profiles of most LNs. An additional reconstruction which closely resembled the cell called 5-HT1 (*Tanaka et al., 2012*), and was neither an ORN, PN nor LN, was full of large dark vesicles (84 ± 12 nm in diameter) and had few synapses with what appear to be small clear vesicles (*Figure 3B–D*) not obviously visible after FIB-SEM imaging. Its axon exited via the mediolateral tract (mlT). Spinules (*Gruber et al., 2018*), interesting membrane invaginations visible in most cell profiles, fall in the vicinity of synapses, but were neither invariably close nor obviously related to

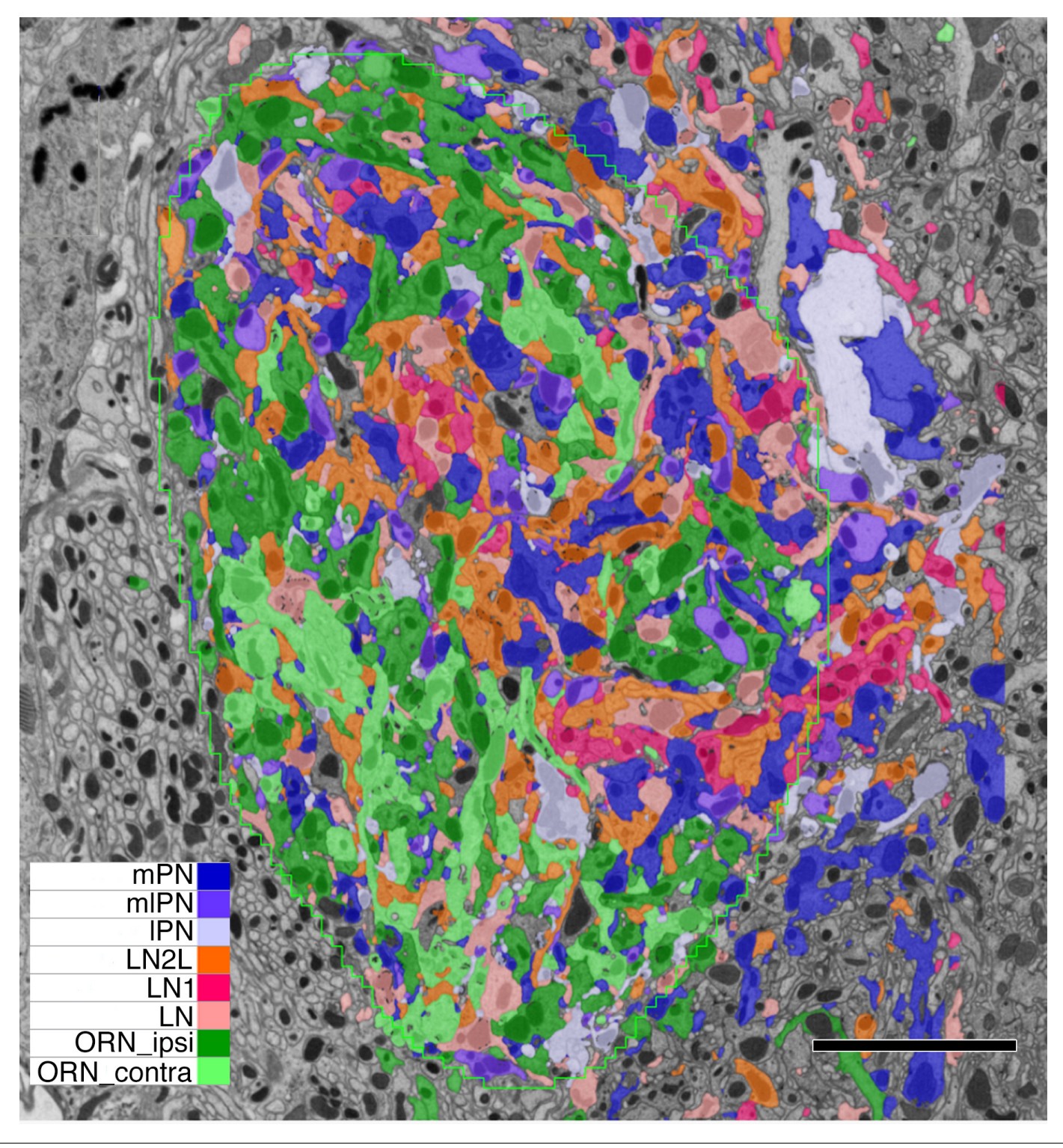

**Figure 2.** Single grey-scale image of the densely reconstructed glomerulus VA1v with colour-coded profiles of different cells (key). Confirming the denseness of reconstruction, note that essentially all profiles are labelled, the few remaining dark grey belonging to orphan elements. Most profiles fall in the range of 0-5-1.5μm in diameter, with larger diameter axons giving rise to tiny dendritic neurites. Scale bar: 5 μm.

DOI: https://doi.org/10.7554/eLife.37550.004

The following source data is available for figure 2:

**Source data 1.** List of quantitative features for all cells of the dataset.

DOI: https://doi.org/10.7554/eLife.37550.005

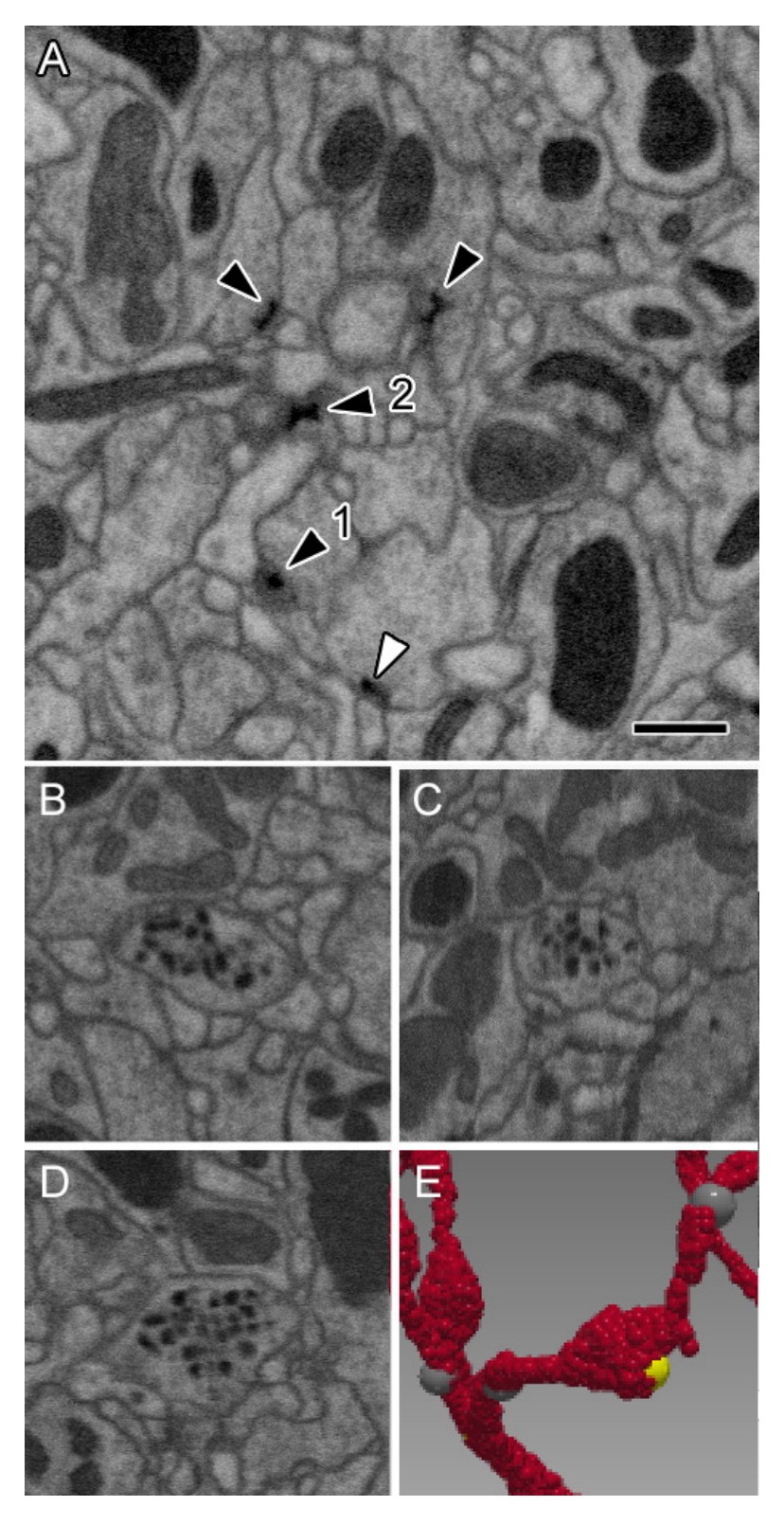

**Figure 3.** Representative synaptic profiles seen in FIB-SEM at 8 nm resolution. (**A**) Five electron-dense profiles of presynaptic sites, four in ORNs and one (open arrowhead) in an LN, reveal a range of shapes, from a clear T-shape, in canonical cross section (1), to cruciform (2), in an en face view of the pedestal, with a range of other profiles that cut the organelle in different planes. Unifying their common identity, all have the same electron density that is clearly visible after FIB-SEM imaging. Postsynaptic densities are not well resolved. The profiles exhibit a wide range of shapes because the neurites

*Figure 3 continued on next page*

*Figure 3 continued*

contributing them are not aligned, as are the columns of the medulla (*Takemura et al., 2015*) and mushroom body output lobes (*Takemura et al., 2017*), and other neuropiles analysed with this imaging method. (**B–D**) Single neurite profile with ~84 nm diameter dense-core vesicles (dcv) viewed in three orthogonal planes, revealing each dcv as approximately circular in all three planes, and thus a sphere. (**E**) Same neurite reconstructed to show a varicosity with a single presynaptic site (yellow). Scale bars: 500 nm.
DOI: https://doi.org/10.7554/eLife.37550.006

synapses; they may represent regions of temporary membrane disequilibrium at sites of vesicle exo- or endocytosis.

## Identified neurons from dense reconstruction in VA1v

We reconstructed 192 neuron bodies (*Figure 4*), each of which had $\geq$50 contacts in glomerulus VA1v. These 192 (*Figure 4—source data 1*, *Figure 4—source data 2*, *Figure 4—source data 3*, *Figure 4—source data 4*) constituted most of the neurons of that glomerulus, which was thus densely reconstructed (*Figure 2*). Occasional orphan elements (*Figure 2*; *Figure 4—source data 4*) may have constituted an additional cell but were disconnected and had membranes too obscure to trace with greater certainty. Otherwise dense reconstruction provided reassurance that no cell could hide undetected within the neuropile (*Figure 2*). A total of 87.9% of VA1v synaptic connections have been assigned to these neurons, the remainder being those of orphan neurites or neurons with fewer than 50 contacts. Overall, glomerulus VA1v had a neuropile volume of 4,858 $\mu m^3$ and the 192 neuron bodies constituted 87% of the volume. Neurons were additionally traced sparsely beyond the borders of the glomerulus where possible, in order to connect to cell bodies, and project via the three ALT tracts, antennal nerve, and commissural tract. The total neurite length traced was 15.8 cm.

We reconstructed the neurons and compared their reconstructed shapes within the glomerulus, and their wider axon trajectories within the antennal lobe, with those revealed by comprehensive mapping using Gal4 enhancer-trap strains (*Tanaka et al., 2012*). This enabled us to identify them cell-by-cell as one of the three main classes of neurons: input ORNs, output PNs, and local interneuron LNs (*Figure 4*). The provisional identification of cells relative to published trajectories of cells in reporter lines is listed in *Table 1*.

A total of 51 ORNs originated in the ipsilateral antennal nerve with a further 56 that entered in the commissure from the contralateral lobe.

We have identified an unanticipated large number of **18 PNs** that lay within the region containing glomerulus VA1v. They fall into the three groups that exit the three main tracts of the antennal lobe (tract nomenclature from *Ito et al., 2014*): (a) Within the group that exits the medial tract we found three tightly interwoven monoglomerular PNs, the mPN1s (*Figure 4*; *Figure 4—source data 2*). (In a previous study *Yu et al., 2010* identified five mPN cells in VA1lm, which we interpret to reveal the PNs of two glomeruli, one with 3 mPN1 and its neighbour with 2) We also identified three other mPN1s with a few projections within the ROI but these were in two other glomeruli. These are labeled mPN1 external (see *Figure 4—source data 2i–n*). b) In the group that exits the mediolateral tract there was a total of eight neurons: (i) two mlPN1s which are monoglomerular; (ii) three mlPN2s which are multiglomerular but only extending neurites to select glomeruli (one having a less widespread projection pattern in VA1v) and one other, mlPN2, which is mainly external to Va1v; and (iii) two mlPN3s which are multiglomerular and have a much sparser projection. For all of these, see *Figure 4—source data 2o–v*. c) In the group that exits the lateral tract we identified eight neurons: (i) three lPN4s that project mainly to VA1v and its sister glomerulus VA1d; (ii) four lPN2s which are multiglomerular and bilateral; and (iii) one cell called lPN2_Comm which resembles lPNS but does not exit in the lateral tract, instead entering from the commissure (see *Figure 4—source data 2b–h*).

One additional cell, AL-AST3 reported by *Tanaka et al., 2012*, with few pre- and postsynaptic contacts, had an axon that exits at the posterior of the antennal lobe and a cell body close to the antennal nerve.

We also identified no fewer than 56 LNs that innervated our glomerulus, of the >150 previously reported for the entire antennal lobe (*Chou et al., 2010*). As previously reported (*Chou et al., 2010*; *Das et al., 2011*; *Liu and Wilson, 2013*), these are morphologically diverse, possibly reflecting the three developmental modes of their origin, as residual larval LNs, as adult-specific LNs emerging

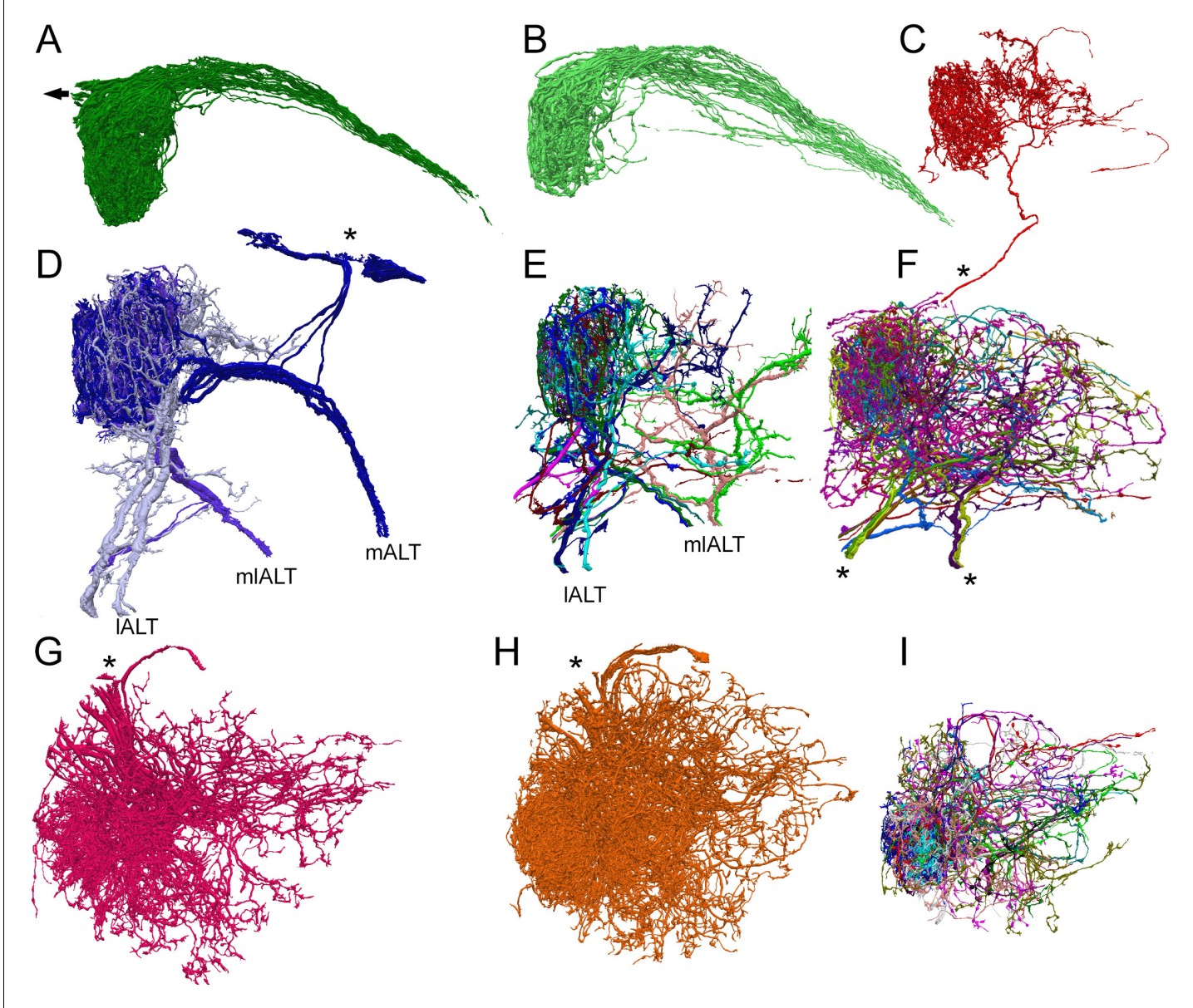

**Figure 4.** Reconstructions of the three main types of antennal lobe neurons in glomerulus VA1v, for comparison with published single-cell reporter expression lines (*Tanaka et al., 2012*). Dorsal (**A–F**) and frontal (**G–I**) views. (**A, B**) Composite of 51 ipsilateral (**A**) and 56 contralateral (**B**) ORNs. Individual cells shown in *Figure 4—source data 4* (see also library of cell types in *Tobin et al., 2017*, their Figure 1—figure supplement 1). Arrow in (**A**) represents neuron path to antennal nerve. (**C**) A single anomalous LN (possibly LN2V in *Figure 2B* in *Tanaka et al., 2012*). Asterisk shows path to cell body beyond region of segmentation. (**D**) The three mPN1 (dark purple) are monoglomerular in Va1v with axons that exit via the medial tract mALT (cf *Figure 3A* in *Tanaka et al., 2012*). Soma locations marked with an asterisk. Two monoglomerular mlPN1 cells (intermediate purple) with axons that exit via the mediolateral tract mlALT (cf *Figure 4A* in *Tanaka et al., 2012*). Three lPN4 (light purple) that arborize in Va1v and its sister glomerulus Va1d, have axons that exit via the lateral tract lALT (cf *Figure 6D2* in *Tanaka et al., 2012*). (**E**) Additional multiglomerular PN types. For all PNs see *Figure 4—source data 2*. (**G**) 17 LN1 cells with somata in the dorsolateral cortex (cf *Figure 2A* in *Tanaka et al., 2012*). (**H**) 24 LN2L cells (cf *Figure 2B* in *Tanaka et al., 2012*). (**F, I**) Composite images of other cells identified as LNs of types LN3-6 and six multiglomerular local interneurons (LNs) differ in the position of their somata and the incoming tract of their axon; these are not illustrated by *Tanaka et al., 2012*. For a library of densely reconstructed cell types that arborize in VA1v, see , *Figure 4—source data 1*, *Figure 4—source data 2*, *Figure 4—source data 3*, *Figure 4—source data 4* ); most are incompletely traced to other glomeruli.

DOI: https://doi.org/10.7554/eLife.37550.007

The following source data is available for figure 4:

**Source data 1.** Library of reconstructed ORNs.

DOI: https://doi.org/10.7554/eLife.37550.008

*Figure 4 continued on next page*

*Figure 4 continued*

**Source data 2.** Library of reconstructed PNs, some partially so.
DOI: https://doi.org/10.7554/eLife.37550.009
**Source data 3.** Library of partially reconstructed LNs.
DOI: https://doi.org/10.7554/eLife.37550.010
**Source data 4.** Library of other reconstructed cells.
DOI: https://doi.org/10.7554/eLife.37550.011

before associated sensory and projection neurons, and as LNs that emerge after synaptic connections are established (*Liou et al., 2018*). As a whole, we find they have a wide range in the numbers of their synaptic contacts. LNs were traced comprehensively in VA1v and also via their axon to their soma; thus although most are panglomerular (*Chou et al., 2010*), our reconstructed LNs are therefore partial. Their morphologies are illustrated fully in *Figure 4—source data 3*. Within the different LN groups, some have many synaptic contacts within VA1v. We could assign most to the six morphological classes identified by *Tanaka et al., 2012*, as follows: (a) The most noteworthy LN is a single largely uniglomerular cell that we identified as LN2V. Although superficially resembling a PN, it does not project to any of the three tracts and has a small neurite which extends towards the ventral cortex of cell bodies but which extends beyond the field of view. This is remarkable in three ways that distinguish it from multiglomerular LNs: in arborizing predominantly within a single glomerulus, in being highly directed (with predominant output to PNs), and in the numbers of its synaptic contacts in glomerulus VA1v. Multiglomerular LNs (*Tanaka et al., 2012*), by contrast, all have fewer synaptic contacts and lack clear directionality.From their morphology and connectivity, we could recognise at least five other types of LNs: (b) Two major groups previously reported are LN1 and LN2L, which share the same soma locations and enter by the same tract. We distinguish them by whether they made contact with ORNs (LN2L: having 24 cells) or not (LN1: having 17 cells) (*Figure 5—figure supplement 1*). LN3 (one cell) and LN4 (one cell) both have somata ventrolateral to the antennal lobe and project to the commissure, LN4 arborizing in only a few glomeruli, and LN6 (two cells each) have few connections in VA1v. In addition a further four bilateral cells send neurites into VA1v via the commissure; their somata are located on the contralateral side. These types are hard to differentiate further without knowing their contralateral arbor. (c) We found in addition LNV (three cells) and LN_LV (three cells), were not easily reconciled with the report of *Tanaka et al., 2012*. The cells called LNV enter the antennal lobe close to the lateral tract and branch profusely; we could not identify these further because we could not find their cell bodies. Three cells we call LN_LV have axons that exit close to the lateral tract and originate from cell bodies close to the antennal nerve; one has an identified axon to the commissure (*Figure 4—source data 3*) and could therefore be LN3. We could not find the tract to the commissure to know whether this cell might be bilateral. d) Four cells with axons that arrive or exit in the commissure are LN commissure cells. These could also be PNs.

Most cells formed both pre- and postsynaptic contacts (*Figure 6*). However, ORNs were predominantly presynaptic, ipsilateral ORNs carrying more synapses than contralateral ORNs, and most of the reconstructed neurites from VA1v PNs were predominantly postsynaptic, as also recently reported from ssEM for other glomeruli (*Rybak et al., 2016*; *Tobin et al., 2017*)(*Figures 5* and *6*). LNs had about equal numbers of both pre- and postsynaptic contacts and, despite some details (*Rybak et al., 2016*), these have not previously been reported comprehensively (*Figure 2—source data 1*). A log/log plot of post- and presynaptic contacts for each cell shows that the predominantly presynaptic ORNs cluster beneath the corresponding values for LNs; having more outputs than inputs (*Figure 6*).

ORN, PN and LN cells exhibit a wide range of synaptic loads, most obvious for the PNs which cluster less tightly, and three of which, all mlPNs, lack all presynaptic sites (*Figure 6*). The three tightly clustered mPN1s by contrast have both pre- and postsynaptic sites, as also reported from ssEM for other glomeruli (*Rybak et al., 2016*; *Tobin et al., 2017*). Means for the numbers of synapses are given for each cell and cell type in *Figure 2—source data 1*.

We can set these features in synaptic organisation in the context of known responses of glomeruli to odours and odour mixtures: Relative to ORN and PN cells, LNs exhibit far greater synaptic

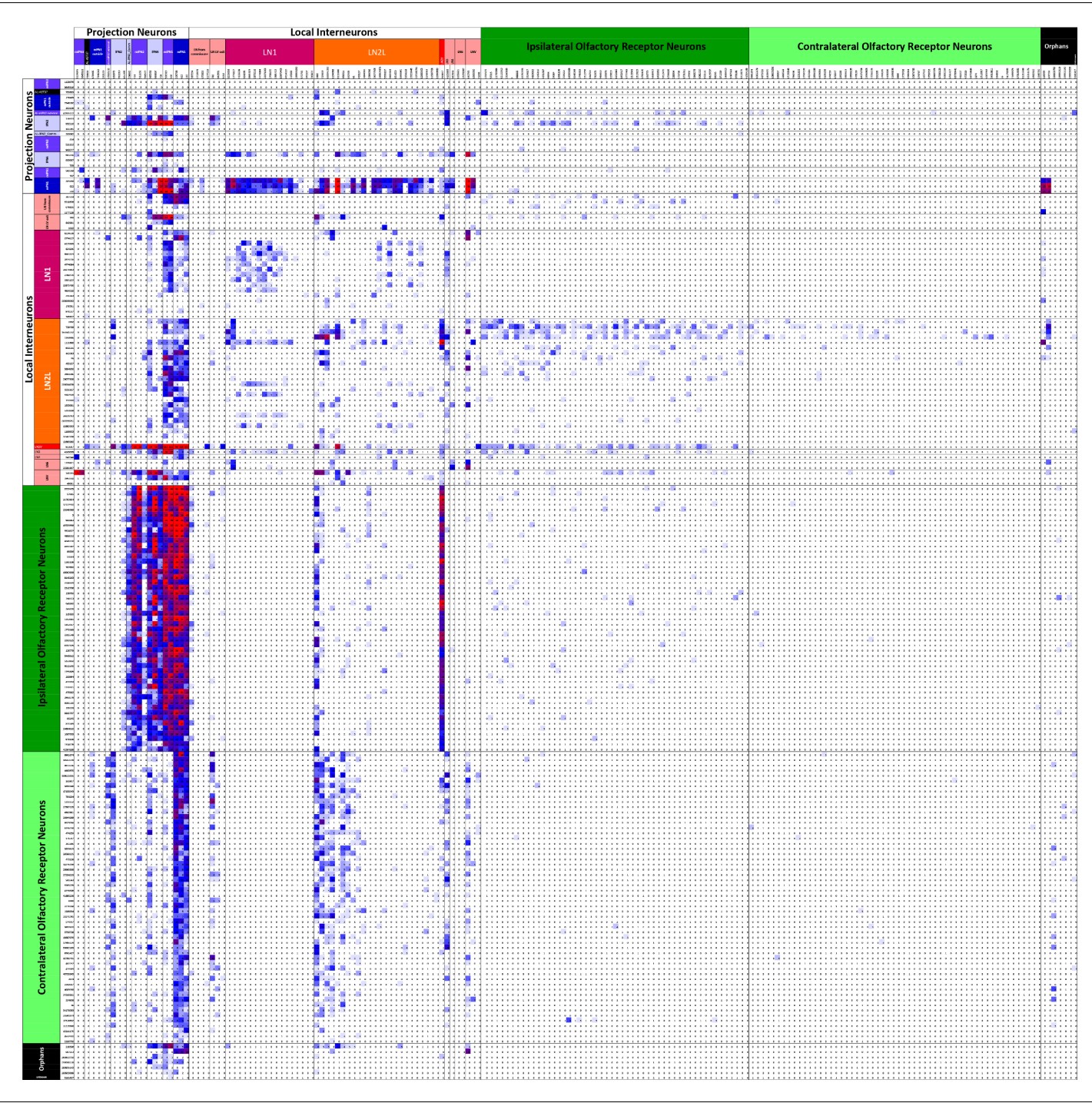

**Figure 5.** Connectivity matrix of VA1v cell types. Register of cells with presynaptic sites (x axis, ordinate) plotted against the same cells having postsynaptic sites and colour-coded intercepts denoting the number of synaptic contacts between each pair of cell types (key), and thus the anatomical strength of their connection. Cells are arranged from the top left origin as, first, outputs (PNs), then interneurons (LNs), and finally inputs (ORNs) and further organized within those groups by the particular cell. Among the total of 192 cells, dense pathways occupy few intercepts, mostly concentrated in ORN to PN, and PN to LN intercepts. Only cells with more than 50 pre- or postsynaptic contacts are included. For the complete matrix, as a spreadsheet, see *Figure 5—source data 1*.

DOI: https://doi.org/10.7554/eLife.37550.012

The following source data and figure supplement are available for figure 5:

**Source data 1.** Connectivity matrix as an Excel spreadsheet file for all 192 antennal lobe glomerulus VA1v cells having >50 contacts.

*Figure 5 continued on next page*

*Figure 5 continued*

DOI: https://doi.org/10.7554/eLife.37550.014

**Figure supplement 1.** Fraction (ordinate, log) of pathway strengths between connected neuron pairs (abscissa, log) from data in the connectivity matrix (*Figure 5—source data 1*).

DOI: https://doi.org/10.7554/eLife.37550.013

diversity (*Seki et al., 2010*). The synaptic summary we present indicates that LNs are anatomically qualified to relay information within and between glomeruli, and given that each glomerulus processes information about only a single olfactant (e.g. *Silbering and Galizia, 2007*), are therefore qualified as a substrate to make comparisons between different olfactants. Comparison between ipsi- and contralateral inputs to PNs may provide a pathway for *Drosophila* to signal the direction of olfactants while PN responses to olfactant mixtures provide evidence for interglomerular inhibition (*Silbering and Galizia, 2007*).

## The synaptic matrix of VA1v

In summary, we first present the numerical features of cells contributing the VA1v connectome. A total of 51 ORNs originated in the ipsilateral antennal nerve with a further 56 that entered in the commissure from the contralateral lobe. Together these innervated 18 PNs, and some of the 56 LNs we found. The latter have mostly not previously been reported at EM level.

We next report the synaptic matrix for 192 neurons having at least 50 connections to other synaptic partners in VA1v. Details of the synapses themselves are reported above and illustrated in *Figure 3*. After manually identifying all synapses and dense proof-reading we found 11,144 presynaptic T-bars opposite which sat 37,843 postsynaptic dendrites, most as triad synapses, with roughly 2.29 synapses per $\mu m^3$. Of these, 93% were resolved with both pre- and postsynaptic partners identified, including some connecting in neighbouring glomeruli.

The synaptic organization has not previously been reported for specific PN types, while the anatomical connections of LNs have not so far been reported at all. Synaptic engagements were specific for each cell type. The complete inventory of chemical synaptic contacts is given as a matrix of their pre- and postsynaptic connections (*Figure 5—figure supplement 1*). Each intercept represents the number of contacts between a single pre- and postsynaptic neuron class, even though individual synapses incorporate several (most frequently three) postsynaptic elements, so that multiple such intercepts are in fact coordinately linked. Indicating the specificity of connections, most intercepts are blank. The matrix in *Figure 5* reports VA1v neurons as classes, and is expanded in spreadsheets to reveal the number of contacts between individual pre- and postsynaptic neurons (*Figure 5—figure supplement 1*). Even though the matrix reveals only the numerical strength of synaptic partnerships and not their physiological strength, it is clear that particular pathways preponderate. These include ORN input to PNs, and selected types of PNs onto some LNs. LN2V, a particular LN considered above, stands out from other LNs in also being highly synaptic, receiving heavy input from ORNs and output to PNs. The matrix we present will serve to interpret future electro- or optophysiological recordings from genetically identified neurons of the glomerulus in terms of both ORN inputs and LN outputs, and especially the connections between LN partners.

The relative frequency of such pathway strengths shows that synaptic load is distributed somewhat unevenly, and that most connections have few synapses, in a distribution that is heavy tailed (*Klebanov, 2003*) (*Figure 5—figure supplement 1*). About 37% of connected cell pairs have three or more connections (see Materials and methods). Some assurance for the accuracy of each intercept in the matrix was also provided by the large number of some partnerships, especially between ORNs and PNs and LN2V, and others, 778 with >10 synapses, relative to all others. The matrix exhibited remarkable selectivity. Thus a total of 36,864 contacts was possible within the matrix of 192 cells, but of these only 8398 pairs (23%) actually made contact, and only 3073 (8%) did so with >2 contacts.

Based on the matrix numbers, the ratio of pre- to post- connections for ORN's was 3.94 for ipsi and 5.29 for contra, while for all PN's it was 1.3, although this includes one outlier lPN2 which had a ratio of 19.5; excluding the latter gave a ratio for PNs of 0.23, while LN1 and LN2Ls had nearly equal numbers of each, with a ratio of 1.09. Ratios for individual cells and averages for groups are given in *Figure 2—source data 1*.

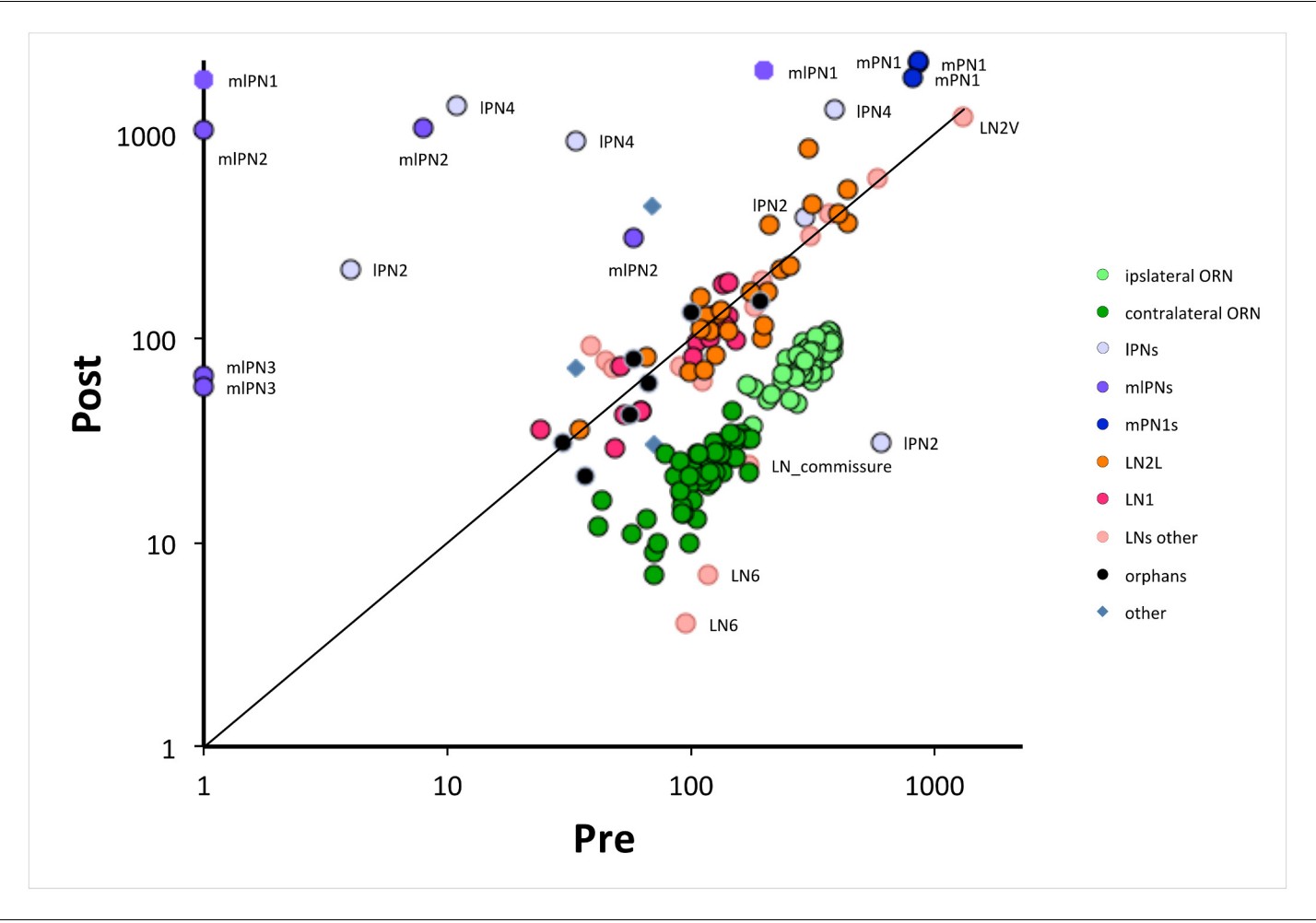

**Figure 6.** The numbers of pre- and postsynaptic sites for each cell, log-log plot. Only cells with >50 contacts are shown. Cells are colour-coded for their type (ORN, LN, PN: key); ORNs are either ipsi or contralateral, PNs are labeled as well as anomalous LNs, which do not cluster with other LNs. Cells can be identified from their pre:post synapse ratio. Almost all LNs have equal numbers of pre and postsynaptic contacts; ORNs have more presynaptic than postsynaptic contacts, with a ratio of 3.9 ± 0.6 for ipsilateral and 5.3 ± 1.5 for contralateral (*Figure 2—source data 1*). PNs are predominately postsynaptic but have more variable pre/post ratios as well as number of contacts than the other two cell types; mPN1s have most presynaptic sites with an average ratio of 0.4 ± 0.03 and mlPN2s are almost completely postsynaptic with a ratio of 0.06 ± 0.11; some PNs have >1000 contacts. Unknown cells, some orphans (not traced to identified neurons), cluster with local neurons, their suspected source. A few cells are anomalous, some (especially LNs) falling outside their cluster are possibly mis-assigned.

DOI: https://doi.org/10.7554/eLife.37550.015

## Connectivity features of the network

If we summarize the numerical features of the network (*Figures 5* and *7*) we see that the output from ORNs numerically dominates the connectome, with 59.8% of all pre- connections, of which 39.5% go to the PN's, 14.5% to LNs, and 5.9% to other ORNs (*Figure 7A*). Much output from the local interneurons is thought to be inhibitory (*Wilson and Laurent, 2005*) and anatomically this output accounts for a further 27.8% of synapses, distributed evenly amongst the three classes of target cells: 10.4% to PNs, 7.8% to ORNs, and 9.6% to other LNs. PN output is rare within the glomerulus, with 6.4% to LNs, 4.1% to PNs and only 1.8% to ORNs, but of course most output is predicted to lie among the target circuits in the mushroom body calyx (*Yasuyama et al., 2002*); *Leiss et al., 2009*; *Butcher et al., 2012*) and higher olfactory centres. Many cells are multiglomerular, and generate a complex interglomerular network inferred mostly from light microscopy and thus not at synapse level. Thus inhibitory LN output to other LNs is predicted to generate disinhibition, and is further expected to constitute part of a widespread weak interglomerular inhibitory network. Glomeruli

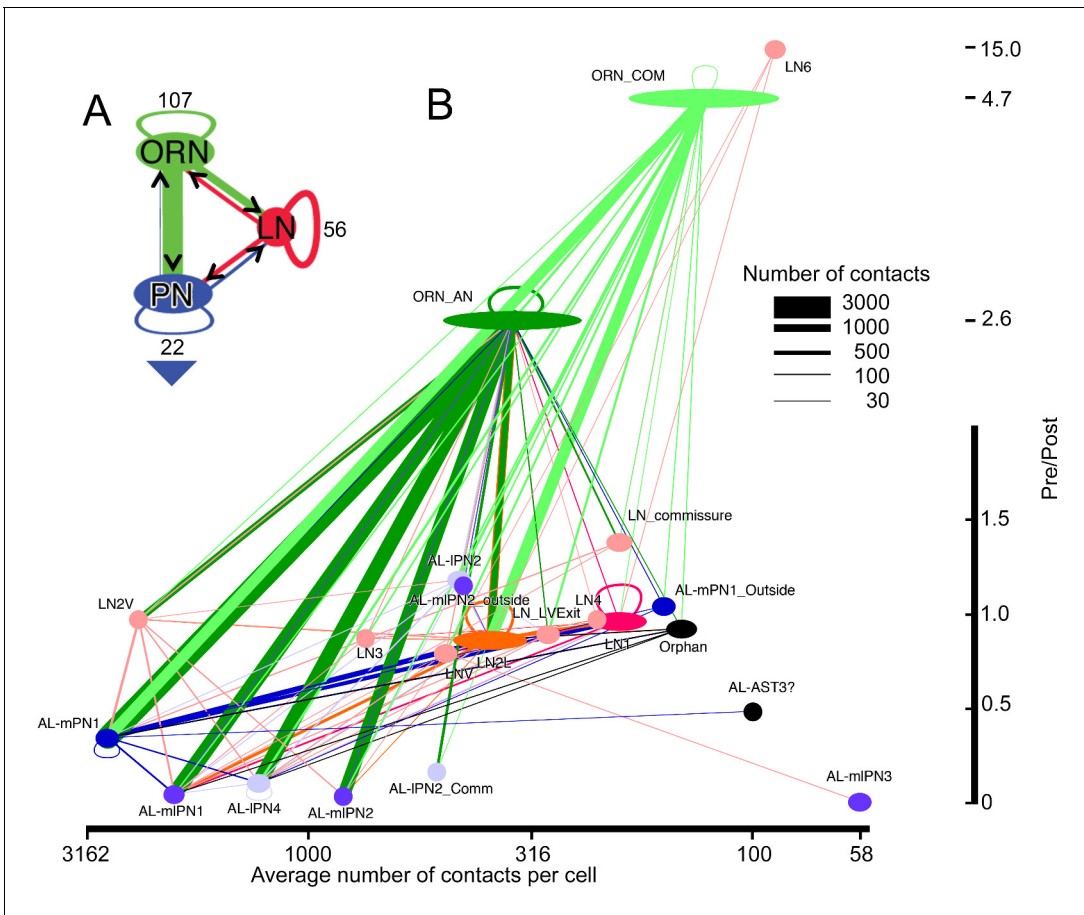

**Figure 7.** Network diagram for all cells and cell types within glomerulus VA1v having >50 pre/post contacts and reported in the overall matrix (*Figure 5*). Edges show connections by classes of cells, not those of their individual constituent neurons. The width of each pathway arrow shows the number of anatomical synaptic contacts (key). (**A**) Aggregate network for three major classes of VA1v cells. The direction of pre- to postsynaptic contact is shown by the arrow and colour of the line connection between synaptic partner cells. ORN inputs to PNs are numerically predominant. Most LNs are predicted to be inhibitory, and provide pathways that integrate both ORN and PN activity; LNs also form their own network. Numbers indicate cells per class. Recurrent loops are feedback between different neurons of the same class. Large arrowhead indicates PN output to higher olfactory centres in the mushroom body and lateral horn. (**B**) Network diagram, as in A, plotted for subgroups of their type (see *Figure 4* and *Figure 4—Source data 4*). Node width represents cell number per group, colour-coded as in *Figure 6*. Vertical: pre-/post- axis, note that this is interrupted at 2.0; horizontal: number of connections per cell plotted on a log scale.

DOI: https://doi.org/10.7554/eLife.37550.016

innervated by OSNs expressing a narrowly tuned olfactory receptor may have more PNs and fewer LNs, suggesting that these glomeruli perform less lateral processing.

Looking at the network's features more specifically, we see that the largest input extends from ORNs, those from ipsilateral ORNs constituting 70% of the total ORN input (*Figure 7B*). Some ORN synapses are made between an ORN and its neighbouring olfactory receptor neurons. Most of the ORN inputs are made to the three mPN1 neurons (25%), however, although input also falls upon other PNs. A total of 76% of ipsilateral ORN input to PNs goes to the three classes of monoglomerular PNs, mPN1 and mlPN1 and lPN4. The latter projects mainly to VA1v but also its sister VA1d glomerulus (*Tanaka et al., 2012*). Of the 76%, 43% goes to mPN1. The only other PN class to receive major input is the multiglomerular mlPN2 which receives 18.5% of the total ipsilateral ORN input. Thus, very little of the ipsilateral ORN input goes to the remaining PNs. (ii) Cell LN2V is anomalous. Its pattern of connections resembles that of a PN more than that of the other LNs, receiving its input from ipsilateral ORNs and providing most of its output to PNs. This pattern does not resemble in all details any of the cells previously reported by *Tanaka et al., 2012*. It appears to be predominantly monoglomerular in VA1v but has sparse projections to other glomeruli. (iii)Unusual among LNs, LN6

is almost entirely presynaptic, with weak output to other LNs and a single strong output to LNv not identified by *Tanaka et al., 2012*. Possibly these two neurons are not LNs at all, but the arbors of receptor neurons from the maxillary palps. Their axons enter the glomerulus from the posterior antennal lobe, compatible with an entry from the antenno-suboesophageal tract, which was not clearly identifiable in our reconstructions; against this possibility, however, ORNs from the maxillary palps are not reported to enter VA1v (*Couto et al., 2005*). (iv) Neurons that project to many glomeruli, as identified from reporters (*Couto et al., 2005*; *Tanaka et al., 2012*), form few connections in VA1v. (v) Six orphans that lack a soma or identified projections outside the glomerulus are included in the matrix. They have very fine neurites we were unable to trace completely, for which reason we have recorded them as orphans, and they are most likely to be the disconnected parts of recorded LNs.

## Synaptic reciprocity

Some connections between neurons are made reciprocally, and many LN cell pairs in particular are so connected, but weakly so (*Figure 8*). The numerical ratios of synapses formed between different cell types differ. Of 1540 possible pairs of neurons having >2 contacts, only 442 (29%) were reciprocal, 380 (25%) of which met a criterion for strong reciprocity in which there were at least four times as many synapses in one direction as the other (a pre:post ratio of either 4 or 0.25: *Figure 8*).

We next consider reciprocity between particular cell pairs (*Figure 8*). There is considerable reciprocity with the two main LN classes (LN1 and LN2L): a) Between types LN1 and mPN1, 74% of pairs are reciprocal with 93% of pairs having a direction of mPN1 (pre) to LN1 (post) and an average ratio of 2.2; and b) between LN2L and mPN1, for which 84% of pairs are reciprocal, with 87% of those pairs being mPN1 (pre) to LN2L (post) and having an average ratio of 1.5. The mPN1 are all highly reciprocal on each other with an average connection strength of 9.9.

LN2L forms many reciprocal connections to other partners, too. Pairs of cells comprising LNs as one partner constitute 85% of the total of all reciprocal cell pairs in the matrix. LN reciprocity is widely observed between many LN pairs but mostly undirected, and with relatively few synapses in either direction. These are likely to be inhibitory (*Wilson and Laurent, 2005*) and feedback from LNs may therefore serve to suppress, or curtail signalling in both LN partners, and thus be disinhibitory, rather than to shorten transmission in the forward direction. Such reciprocal partnerships form connections on average of 6.7 ± 4.8 synapses (8.1 ± 5.2 in the stronger direction and 4.7 ± 3.1 in the weaker). In comparison, the average connection strength of ipsilateral ORNs to mPN1 is 21.6 ± 7.7 and for contralateral ORNs to mPN1 is 9.2 ± 4.6 synapses, compared with a recent report (*Tobin et al., 2017*) which found an average of 23 synapses from ipsilateral ORNs.

## Discussion

Our study reports the complete synaptic connectome for all three cell types, ORN, LN and PN, of a single large glomerulus of the *Drosophila* antennal lobe, providing a proof of principle for the remaining 50 glomeruli. It complements two contemporary EM studies for different glomeruli (*Rybak et al., 2016*; *Tobin et al., 2017*), augmenting these by including important synaptic circuits of the LNs, which had not previously been known in any detail at synaptic level yet which constitute nearly half the cells of the glomerulus. Our data are notable for comprehensively documenting synaptic partnerships, and for having overcome the challenge of reconstructing the dense multidirectional arbors of antennal lobe local neurons, aided by the superior z-axis resolution of FIB-SEM compared with ssEM. Unlike PNs and their ORN inputs, and except LN2V, we find that LNs have three main characteristics: they have fewer synapses on average than PNs and ORNs; these are heterogeneous, and they form undirected, reciprocal synaptic networks which are inhibitory.

The connectome we report is essential to promote functional analyses using genetic dissection methods (*Venken et al., 2011*). In particular it will enable the interpretation of future analyses of network function made possible: first, by imaging methods, either in vitro, especially using genetically encoded calcium (e.g. *Tian et al., 2012*) or voltage (e.g. *Antic et al., 2016*) indicators; or second, during intact behaviour in vivo (e.g. *Grover et al., 2016*), in response to single odours or their combinations (e.g. *Yasuyama et al., 2002*). Our data also support computational approaches to insect olfaction.

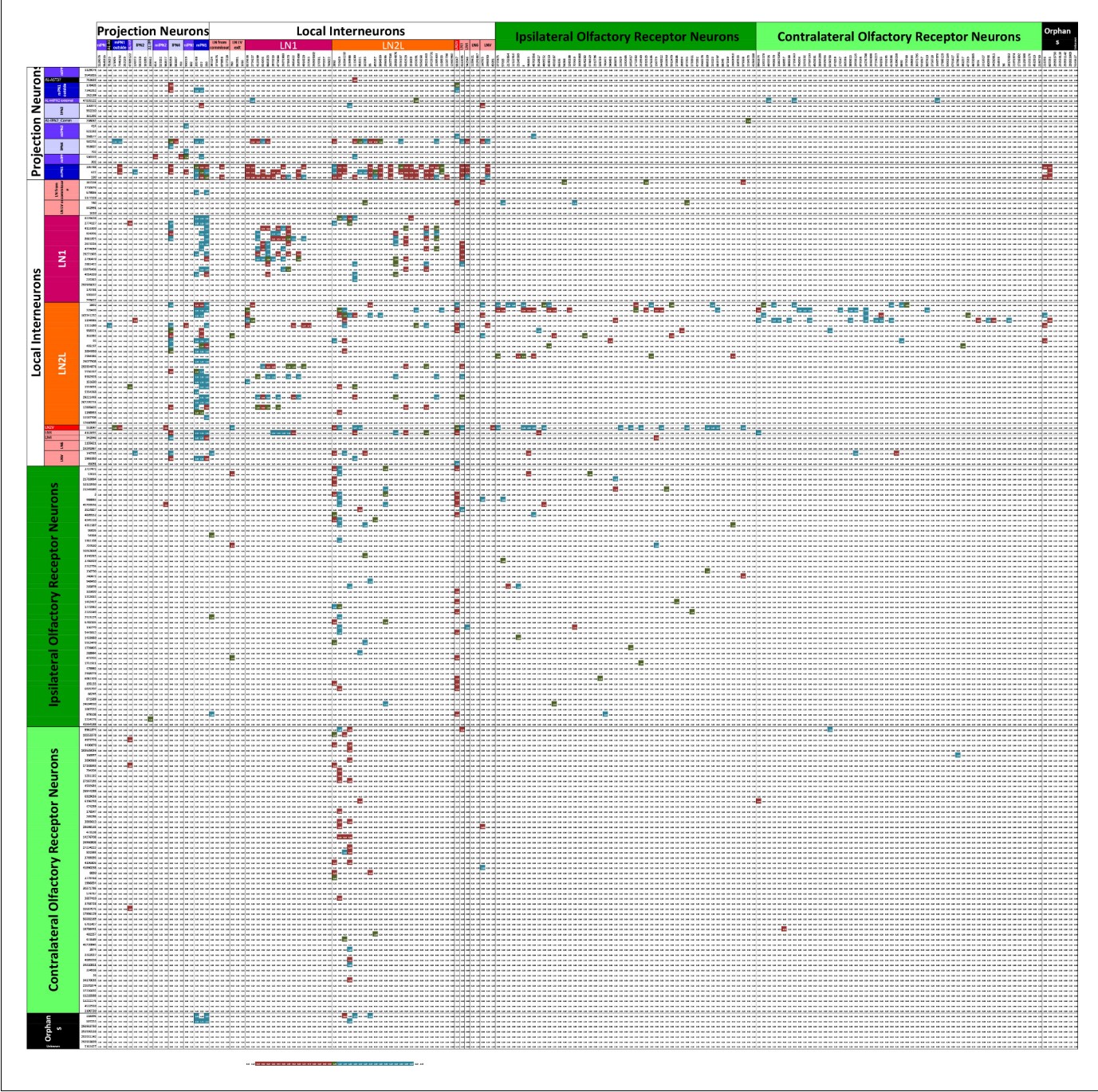

**Figure 8.** Matrix of the 192 cells in *Figure 5* highlighting those pairs that are strongly reciprocal. Pairs have connections of at least three synapses in the pre:post and post:pre directions so having at least six pre- and postsynaptic contacts in total. Each intercept shows the ratio for the pair of partner neurons of its pre- to post- contacts. Ratios of between 0.25 and 4 are highlighted in a graded fashion (see scale). These represent those partner neurons at which the numerical strength of pre and postsynaptic pathways is at least 4:1, where four represents a balance in favour of the presynaptic pathway, 0.25 represents a balance in favour of the postsynaptic pathway, and a value of 1.0 represents equal numerical balance in both directions. Numerical values are reflected about the diagonal of the matrix, changed inversely between the two sectors, 0.25 in one direction corresponding to four in the other.

DOI: https://doi.org/10.7554/eLife.37550.017

The following source data is available for figure 8:

**Source data 1.** Reciprocity matrix for all 192antennal lobeglomerulus VA1v cells having >50 contacts.
DOI: https://doi.org/10.7554/eLife.37550.018

The volumetric packing density of synapses, one site per 2.29 μm$^3$ of neuropile, corresponds approximately to the differently computed values reported from TEM by *Rybak et al. (2016)* for glomerulus VA7, but is less than the density of synapses in the first and second optic neuropiles, the lamina and medulla (*Meinertzhagen, 2014*). We may speculate that the more densely packed visual synapses also employ higher rates of transmitter release and are consequently more energetically demanding, but deeper comparisons are hard to make.

We provide the complete connectome from a sensory neuropile in *Drosophila*, augmenting previous reports on other brain regions, the optic medulla (*Takemura et al., 2015*) and the mushroom body, both the calyx to which the PN axons project (*Butcher et al., 2012*) and their Kenyon cell output neurons in the alpha lobe (*Takemura et al., 2017*). Both the present report and the previous medulla connectome reveal only part of their neuropile as a whole however, for glomerulus VA1v, a single glomerulus in the antennal lobe, and for seven columns only of the optic medulla (*Takemura et al., 2015*). Like the latter, for all it is comprehensive the connectome of this single *fruitless*-positive glomerulus lacks most of the circuits of any of the approximately 50 other glomeruli, even of its neighbours, to which most of the local interneurons, those that are multiglomerular, also extend.

## Dense reconstruction captures known genetic cell types

Along with genetic labelling from single-cell reporter lines, serial-section EM provides one of two canonical means to identify the morphological signature of a neuron (*Meinertzhagen et al., 2009*; *Meinertzhagen, 2018*). The mostly complete agreement we find between the two methods, reported here for FIB-EM and previously from Gal4 enhancer-trap labels viewed by light microscopy (*Tanaka et al., 2012*), provides assurance that both methods reliably succeed in identifying cell types of glomerulus VA1v, each yielding information that complements the other. From FIB-SEM, in particular, we can be sure that all cells are detected and none can hide in the forest of others, and that apart from orphan fragments the portions of each cell within VA1v are completely identified. This has been important in particular to plot the more elusive connections of the numerous LNs, which branch with greater complexity and are harder to identify than the 18 PNs, and which previous accounts lack at a synaptic level. On the other hand, most LNs are multiglomerular (*Chou et al., 2010*; *Tanaka et al., 2012*) so that their arbors in glomeruli other than VA1v still await completion, and we were unable to trace some neurites to a soma, making cell identification problematic in a few cases.

The synaptic connections we report confirm many that have been conjectured from light microscopy (*Tanaka et al., 2012*), even if the latter lacks the resolution to assert the presence of synaptic contacts. Most obvious is the pathway provided by ORN input to the three mPN1s, which project via the medial antennal lobe tract and provide input in turn to the mushroom body calyx and lateral horn (*Stocker et al., 1990*; *Tanaka et al., 2012*). Reporter Gal4 screens may not have identified all neurons per identified class, however. In particular, from Gal4 lines *Tobin et al. (2017)* report three PNs on the left side glomerulus DM6 and two on the right side, whereas a total of only 59 types for all 50 glomeruli have been labeled by Gal4 drivers in another study (*Tanaka et al., 2012*), an average of about one PN per glomerulus. In a more complete inventory revealed by photoactivating GH146-Gal4 positive PNs, most glomeruli are reported to contain an average of 2 ± 1 uniglomerular PNs, while six remaining glomeruli, which include VA1v, are innervated by 6 ± 2 PNs (*Grabe et al., 2016*).

## Cellular organization of VA1v

The VA1v glomerulus is innervated by sensilla containing olfactory receptors involved in courtship and mating (*Grabe et al., 2016*) and additionally exhibits sexual dimorphism (*Kondoh et al., 2003*; *Stockinger et al., 2005*), its output delivered by the three uniglomerular mPN1s that project to the mushroom body and lateral horn (*Tanaka et al., 2012*). These possibly utilize a strategy to ensure that highly essential odor cues are transferred reliably and quickly from the antennal lobe to higher brain centers. Coincidentally they offer the eventual prospect to examine the network basis for sexual dimorphism in the antennal lobe, a prospect made possible by the connectome we report here.

The large number of VA1v PNs is a feature predicted to improve the detection accuracy and latency of odor stimuli (*Jeanne and Wilson, 2015*) and may also encode diverse stimulus features (*Grabe et al., 2016*). The large number leads us to anticipate that the number of LN dendrites is, by

contrast, small relative to other glomeruli (*Grabe et al., 2016*), but we lack data for the latter. The number of PNs previously seen in genetic screens is significantly fewer than the 18 we find from FIB-SEM, but distinction between mono- and multiglomerular PNs must be made for valid numerical comparisons. Even so, the reasons for such wide differences are not entirely clear but suggest that the genetic methods so far used underestimate the reported PN numbers, and thus that GH146 Gal4 screens are far from saturated. This is a major conclusion to be drawn from our connectome.

In addition to their number, or possibly as their correlate, the PNs of VA1v as a whole exhibit considerable morphological diversity between classes. This is probably not typical of most antennal lobe glomeruli that have fewer PNs, as for example is clearly the case for glomerulus DM6 for which the monoglomerular PN synaptic connections have also been recently reported (*Tobin et al., 2017*). Supporting the uniformity of DM6 PNs as a class, pairs of PNs simultaneously recorded in the same glomerulus have well correlated levels of neuronal activity (*Kazama and Wilson, 2009*). Morphological diversity cannot of course even be anticipated for the many other glomeruli that contain only a single PN (*Grabe et al., 2016*). The situation remains open for yet other glomeruli and VA1v is probably not unique in containing so many PN types. For example, DA1 with at least 9 PNs (*Grabe et al., 2016*) contains PNs with different origins, from both the lateral (*Lai et al., 2008*) and anterodorsal (*Lin et al., 2010*) lineages. The latter generates exclusively uniglomerular PNs that project through the inner antennocerebral tract, while the lateral lineage generates various types of neurons, including uniglomeurular PNs (*Lin et al., 2010*).

## Synaptic identification and composition

We have annotated contacts with anatomical features of chemical synapses, in particular with T-bar ribbons and presynaptic vesicles. As for other connectomes, in the antennal lobe (*Rybak et al., 2016*; *Tobin et al., 2017*) and optic neuropiles (*Takemura et al., 2013*; *Takemura et al., 2015*), we were therefore unable to annotate appositions that we could reliably interpret as putative gap junctions (*Bennett and Goodenough, 1978*) between neurons. The tortuosity of the tightly woven neurites prevented us from visualising linear stretches of membrane with close appositions, having densities on the membranes of both sides, that might have provided evidence for candidate gap junctions, like those reported from more favourable sites (e.g. *Shaw and Stowe, 1982*).

Our findings clearly establish that each of the three major classes of neurons is both pre- and postsynaptic to the other two, that in general most neurites have an approximate ratio of one presynaptic site for three postsynaptic contacts, most synapses thus being triads or tetrads.

## Neurotransmitters

Detailed evidence for the neurotransmitters used by identified antennal lobe neurons is far from complete. As single classes, the ORNs express a cholinergic phenotype (*Wilson, 2013*), as do the projection neurons, at least those that project to the mushroom body calyx (*Yasuyama et al., 2002*) and lateral horn (*Yasuyama et al., 2003*), while LNs are likely to be either GABAergic (*Wilson and Laurent, 2005*; *Okada et al., 2009* ; *Seki et al., 2010*), and thus inhibitory, or glutamatergic and thus probably also inhibitory (*Liu and Wilson, 2013*).

## Synaptic reciprocity

One opportunity enabled by analysing an entire connectome is to identify comprehensively the extent of synaptic reciprocity between neuron partners. Widespread synaptic reciprocity has been identified among synaptic circuits in many different brains and is hardly new. Various forms have in fact been identified and named, mostly for their inferred or demonstrated functions (*Shepherd, 1998*). Indeed, the anatomical demonstration of reciprocal synapses was reported in very early EM studies, both on the vertebrate olfactory bulb (*Rall et al., 1966*), and among the circuits of the inner retina, at feedback synapses of amacrine cells (*Dowling and Boycott, 1966*). But its wider documentation has mostly lacked specific attention or quantification and the full extent of synaptic reciprocity has never been fully revealed, in a way now possible from recently reported entire connectomes. Not in fact since such data have become available has it been possible to examine this question further, as is now possible in *Drosophila*, in the lamina (*Zhao et al., 2015*), medulla (*Takemura et al., 2015*), and mushroom body (*Takemura et al., 2017*). The optic neuropiles in *Drosophila* reveal the widespread incidence of synaptic reciprocity among visual circuits, while in the

tadpole larva of *Ciona* the total proportion of reciprocal synaptic connections between neuron pairs is 0.39 (*Ryan et al., 2016*). By comparison we find a ratio of 0.25 for strongly reciprocal pairs in the antennal lobe. *Takemura et al., 2015* discuss the possibility that the reciprocity of synaptic connections may be used to offset variation in the number of synapses in either direction, the operation of a synaptic circuits then depending not on the strength of a particular connection but on its ratio with other connections. Implicit in the reciprocal arrangement of synaptic partners is the fact that transmission in forward and backward directions should have opposite polarities.

In contrast to the ORNs and PNs, LN input to other cells although widespread, exhibits a wide range of synaptic connections that are numerically weak, overshadowed by the numerical strength of ORN input to PNs. Some LNs share reciprocity with a multiglomerular LN partner, as part of a weak lateral interglomerular network that is presumed to be mostly inhibitory. Our findings highlight in particular the extent of reciprocity between the synaptic elements of VA1v and the undirected nature of LN networks, the function of which may be estimated by those that express a GABA phenotype (*Wilson and Laurent, 2005*) or GABA receptors (*Okada et al., 2009*) and are thus predicted to mediate inhibition. The expression of other neurotransmitters offers ground for claims that are less certain.

An exception to these features is shown by LN2V, which is strongly connected to both ORN and PN neurons. The numbers of synapses are otherwise in most cases small, generally not more than 6, compared for example with the ipsilateral ORN to LN2V (about 25 synapses per ORN) and to individual PN pathways (about 100 synapses). Perhaps LN2V is a more direct feedback neuron than other LNs. For further predictions, it will be important to demonstrate that the anatomical ratios for synaptic feedback match the corresponding synaptic currents, which will require identification of postsynaptic receptors and their corresponding membrane conductances. Further comparisons between the connectivity of VA1v and that in other olfactory glomeruli will of course be instructive.

Further predictions for LN function based on our circuit information are still hard to make. It is particularly telling that silencing the populous LN1 and LN2 cells in the entire antennal lobe fails to modulate either glomerular input or output activity, suggesting their lack of influence on odor identity coding as a whole, and their possible role instead in more local interactions, possibly on a global or longer time scale (*Strube-Bloss et al., 2017*), predictions that are currently hard to evaluate anatomically.

## Materials and methods

### Animals

The dissected brain of a 6 day female fruit fly, *Drosophila melanogaster*, a cross between homozygous $w^{1118}$ and CS wild type, was prepared by high-pressure freezing and freeze substitution as previously reported (*Takemura et al., 2013*). For this, the fly was stabilized in a collar (*Heisenberg and Böhl, 1979*), and a 240 μm slice cut from the head in a frontal plane using a vibrating microtome (Leica VT1000), and prefixed for 20 min in 2.5% each of paraformaldehyde and glutaraldehyde in 0.1M cacodylate buffer, then high-pressure frozen and freeze-substituted in 1% $OsO_4$, 0.2% uranyl acetate, 3% water in acetone, and then embedded in Durcupan, as previously reported (*Zhao et al., 2015*; *Xu et al., 2017*).

### FIB-SEM imaging

The right side was trimmed for FIB milling after it had been screened using X-ray imaging (Xradia Versa XRM-510) and the strongly X-ray-positive images used to select the antennal lobe. This X-ray imaging procedure was required to provide the precise depth to enable precise trimming at which FIB imaging should start.

An image stack of ~8,900 FIB-SEM images was then acquired from the block face (*Knott et al., 2008*; *Xu et al., 2017*), collecting images from the brain's right side at a resolution of 8 nm/pixel, in a frontal plane from anterior to posterior, using a Zeiss NVision (*Xu et al., 2017*). Imaging extended for most of the entire antennal lobe's depth and included a portion of the antennal nerve. In between images a focused gallium ion beam was used to remove 2 nm from the block face, and secondary electrons emitted from the block face collected in consecutive images. Minor shifts in these

were aligned using affine transforms and consecutive sets of four images summed to generate 8 nm voxels that after final alignment yielded an isotropic stack with 8 × 8 × 8 nm resolution.

## Glomerulus VA1v

We chose VA1v because it lay close to the lateral border of the lobe, near the entry point of the antennal nerve. The borders between many glomeruli were usually not well distinguished in single images from material prepared after rapid freezing preparation for FIB-SEM imaging (*Figure 2*) which, unlike conventional fixation for ssEM, left glial boundaries expanded and pale. To demarcate the specific glomerulus we first used an algorithm that identified the presynaptic organelles, T-bar ribbons (*Fröhlich, 1985*; *Hamanaka and Meinertzhagen, 2010*), to enable large-scale, automated imaging of synaptic contacts (*Huang and Plaza, 2014*; *Kreshuk et al., 2011*). Applied to the entire antennal lobe's FIB image stack, these generated a synapse point cloud of over 500,000 synaptic puncta (*Figure 1A,B*), as previously reported (*Zhao et al., 2015*). The local variations in the density of synapses, elevated in regions (*Figure 1A*) corresponding to individual glomeruli (*Zhao et al., 2015*), mirrored reconstructions from light microscopy of synapses made visible by immunolabelling the presynaptic protein Bruchpilot (*Laissue et al., 1999*). Detailed comparisons between the light and FIB-SEM datasets enabled us to identify VA1v (*Couto et al., 2005*; *Table 1*). Using these features, a specific region of interest (ROI) was determined using the software tool Neutu. Different Gal4 lines combine the previously reported VA1l and VA1m into VA1v (*Couto et al., 2005*; *Endo et al., 2007*), indicating that not all anatomical territories coincide with genetic boundaries.

## Synapse annotation

For synapse annotation two trained proofreaders who had attained a recall proficiency in excess of 85% and a precision of >94.7% were used to annotate presynaptic T-bars in the EM volume. They marked pre- and postsynaptic sites manually in software, Raveler (https://openwiki.janelia.org/wiki/display/flyem/Raveler) or the later NeuTu (https://github.com/janelia-flyem/NeuTu; *Zhao et al., 2018*) in combination with the Distributed, Versioned, Image-Oriented Dataservice DVID (https://github.com/janelia-flyem/dvid *Katz and Plaza, 2018*). Unlike neuropiles in other species, in *Drosophila* the presynaptic sites are far more easily recognized than the corresponding postsynaptic densities (PSDs). T-bar annotation was undertaken first followed by PSD annotation, so that each T-bar was reviewed again. A small minority of T-bars (<1%) that lacked clear PSDs were removed from the dataset. PSDs were further refined by checking all autapses. A total of 3822 pathways with one synaptic contact were identified. These were all checked by a second proof-reader and approximately 2881 were verified, to confirm both the contact and the neurons that this connects. Of these, only 58 were then denied.

## Reconstructions

Proofreading was performed so that bodies with pre- and postsynaptic partners were connected to progressively more extensive neurites, in both Raveler and NeuTu-EM and in combination with DVID software, according to previously published methods (see *Chklovskii et al., 2010*; *Plaza et al., 2014*; *Takemura et al., 2015*; *Takemura et al., 2017*) by previously trained proof-readers. The image stack was automatically segmented and putative neurite profiles generated (*Chklovskii et al., 2010*; *Plaza et al., 2014*). Reconstructions were made initially by marking pre- and postsynaptic sites manually. Segmentation was undertaken using a context-aware two-stage agglomeration framework (*Parag et al., 2015*). We undertook dense reconstruction of all elements within the glomerulus using focused proof-reading (*Plaza et al., 2012*) and sparse reconstruction beyond its borders to determine the location of somata and axon tracts. The density of connectome reconstruction helped us eliminate errors of omission, instances of failures to detect synapses. Neuron reconstructions were closely examined for proof-reading errors. Consensus-based proofreading using up to four proof-readers arbitrated disagreements in the shapes or connectivities of reconstructed neurons. Proof-reading required us to split reconstructed bodies frequently. Cell types of reconstructed neurons were identified by comparing the shapes of their arbors and by the locations of somata and axon tracts, with those previously reported from light microscopy (*Tanaka et al., 2012*). Connectivity matrices were generated by combining the results of neuron reconstructions with those for the connection strength from the number of pre-/post synapses. Synapses had a multiple-contact

composition, incorporating several postsynaptic neurites linked coordinately to receive transmitter released at a single presynaptic site. A matrix was constructed for all bodies with partnerships exceeding 50 connections between a single pre- and postsynaptic contact. In fact we found three or more connections for about 37% of connected cell pairs, and took this criterion as a lower limit to acknowledge a pathway, disregarding connections with yet fewer examples as possible errors of human provenance caused by incorrect proof-reading. Networks were plotted in Cytoscape, v. 3.5 (*Shannon et al., 2003*).

Our analysis took 60 person months of proofreading time and 20 person months for curation.

## Acknowledgements

This work has been supported by FlyEM at Janelia Research Campus of Howard Hughes Medical Institute. We acknowledge the generous support and encouragement of Gerald Rubin and the following people also at Janelia; in particular Gary Huang, William Katz and Toufiq Parag, Lowell Umayam and Ting Zhao for IT support; and Pat Rivlin and Shinya Takemura for help with glomerulus identification. We acknowledge Nobuaki Tanaka (Hokkaido University, Sapporo) for help identifying cell types; and Mss Aya Shinomiya, Dorota Tarnogorska and Jola Borycz (Dalhousie) for additional proof-reading, and Asa Barth-Maron (Harvard University, Boston) for help with data analysis.

## Additional information

### Funding

| Funder | Grant reference number | Author |
|---|---|---|
| Howard Hughes Medical Institute | Janelia FlyEM | Jane Anne Horne<br>Carlie Langille<br>Sari McLin<br>Meagan Wiederman<br>Zhiyuan Lu<br>C Shan Xu<br>Stephen M Plaza<br>Louis K Scheffer<br>Harald F Hess<br>Ian A Meinertzhagen |

The funder (HHMI) provided technical support for study design, and data collection.

### Author contributions

Jane Anne Horne, Data curation, Software, Formal analysis, Supervision, Methodology, Writing—original draft, Project administration, Writing—review and editing; Carlie Langille, Sari McLin, Meagan Wiederman, Data curation, Validation; Zhiyuan Lu, Validation, Methodology, Ultramicrotomy; C Shan Xu, Harald F Hess, Visualization, Methodology; Stephen M Plaza, Resources, Software, Methodology; Louis K Scheffer, Software, Methodology; Ian A Meinertzhagen, Writing—original draft, Writing—review and editing

### Author ORCIDs

Jane Anne Horne (ID) http://orcid.org/0000-0001-9673-2692
C Shan Xu (ID) http://orcid.org/0000-0002-8564-7836
Louis K Scheffer (ID) http://orcid.org/0000-0002-3289-6564
Ian A Meinertzhagen (ID) http://orcid.org/0000-0002-6578-4526

### Decision letter and Author response

Decision letter https://doi.org/10.7554/eLife.37550.023
Author response https://doi.org/10.7554/eLife.37550.024

# Additional files

## Supplementary files
• Transparent reporting form
DOI: https://doi.org/10.7554/eLife.37550.019

## Data availability

All data generated or analysed during this study are included in the manuscript and supporting files. Source data files have been provided for Figures 5, 8 and Figure 2-source data 1. Grayscale and segmentation data are hosted at a Janelia website: http://emdata.janelia.org/AL-VA1v. Data can be viewed in a web browser using neuroglancer. Please see the readme file on how to access the data programmatically using dvid and DICED (this can be accessed by clicking on "AL-VA1v" (hyperlinked) at http://emdata.janelia.org/AL-VA1v).

The following dataset was generated:

| Author(s) | Year | Dataset title | Dataset URL | Database and Identifier |
|---|---|---|---|---|
| Horne JA, Langille C, McLin A, Wiederman M, Lu Z, Xu CS, Plaza SM, Scheffer L, Hess HF, Meinertzhagen IA | 2018 | Greyscale and segmentation data | http://emdata.janelia.org/AL-VA1v | FlyEM, AL-VA1v |

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
