## [Decision Letter]

Thank you for submitting your article "The complete connectome of a *Drosophila* antennal lobe glomerulus" for consideration by *eLife*. Your article has been reviewed by three peer reviewers, including Liqun Luo as the Reviewing Editor and Reviewer #1, and the evaluation has been overseen by Eve Marder as the Senior Editor. The following individual involved in review of your submission has agreed to reveal their identity: Moritz Helmstaedter (Reviewer #2).

The reviewers have discussed the reviews with one another and the Reviewing Editor has drafted this decision to help you prepare a revised submission.

Essential revisions:

– Please submit as a Tools and Resource paper (given reviewers' reservations about new biological insight)

– Please provide synaptic calibration data (report precision/recall numbers on what they call a synapse) and other relevant metrics

– Please ensure full data availability, which will be checked by reviewers/editors before final acceptance

– Please re-write the manuscript with the general readership of *eLife* in mind, integrating as much as possible previous data on the subject and providing biological significance of the finding.

– Please also address other comments from the reviewers; see below for the full reviews.

Reviewer #1:

Horne et al. reported the reconstruction of the connectome for a VA1v glomerulus of the *Drosophila* antennal lobe using FIB-SIM, a state-of-the-art EM reconstruction method that can achieve 8-nm isotropic resolution. The major advance compared to previous efforts in EM-based reconstruction of antennal lobe connectivity (Tobin et al., 2017; Rybak et al., 2016) is the inclusion of local interneurons (LNs), which are in large numbers and are heterogeneous based on light microscopic single-cell tracing data. Indeed, the authors identified 56 LNs in the VA1v glomerulus, and reported synaptic connections between 56 LNs, >100 olfactory receptor neurons (ORNs), and 18 second-order projection neurons (PNs) in this glomerulus. These quantitative data (summarized in Figures 5 and 7) will be highly valuable in understanding how the *Drosophila* antennal lobe processes olfactory information. However, the paper contains a number of problems that need to be extensively revised before it can reach the high standard of an *eLife* publication.

1) My major critique is that the current writing style makes it very difficult for readers besides a small group of specialists to be interested in reading the paper. The description of the data is accompanied with little biological meanings or insights. While EM reconstruction must describe dry data, this one is an extreme; for comparison, Tobin et al., 2016 is much more interesting to read even for a specialist. Major efforts need to be put to streamline the writing and interpret the data with biological context, with the general readership of *eLife* in mind.

2) There are a few significant errors or vague statements that further impede reading. The most important one occurs in paragraph three of subsection “Identified neurons and their synaptic networks”. I believe "medial lateral tract" should be "medial tract", or mALT. The medial lateral tract contains entirely different type of neurons (belonging to group b). Furthermore, in the same paragraph, it promises three groups, but I did not see c) after a) and b).

3) Assuming my interpretation above is correct, there are only 3 uniglomerular PNs that belong to the mALT group. This number is different from the reported 5 based on lineage tracing using dual-color MARCM (Yu et al., PLoS Biology 2010). The authors should comment on this. In general, it will be useful for the authors to systematically specify which neurons in their reconstruction have been traced to the cell bodies, and what is the location of cell bodies with respect to the antennal lobe neuropil; such information can be presented an additional column or row in Figure 5. These will provide cross-validation as the lineages of antennal lobe neurons have been extensively documented in the literature.

4) Speaking of Figure 5, I need to amplify to about 600% on my large computer screen in order to see the data. I expect that it will be quite difficult to fit into a regular *eLife* page with sufficient resolution. Figure 8 suffers from the same problem. The authors should think of creative ways of presenting these data that are more readily seen by readers.

5) The manuscript contains a number of citation errors. Two examples: Introduction paragraph four, they should cite Marin et al., 2002, which was published back-to-back with Wong et al., and which contains several unique single cell tracings that are relevant to this study. In paragraph seven of subsection “Identified neurons and their synaptic networks”, I think they mean Tanaka 2012 (rather than 2010).

6) The acronym FIB-SIM should be spelled out in Abstract at the first mention. The authors should also consider adding "nearly" before "complete" to their title given that there are still orphan profiles (despite the tremendous amount of work!).

Reviewer #2:

The manuscript "The complete connectome of a *Drosophila* antennal lobe glomerulus" by Horne et al. reports the electron-microscopic imaging and dense connectomic reconstruction of the synaptic circuits in a glomerulus of the antennal lobe in *Drosophila melanogaster*.

This is a comprehensive piece of work containing a lot of potentially useful connectomic information.

In my understanding, compared to previous connectomic reconstructions of glomeruli in *Drosophila,* which the authors cite (Rybak et al., Tobin et al.), this study includes the connectivity of local interneurons which were not available in the previous publications.

Since I am neither an expert in the olfactory system nor *Drosophila* neurobiology I will focus my comments on the EM and connectomic reconstruction aspects of this work.

This is a comprehensive piece of data and I commend the authors for this work, but I have a few concerns that in my view are substantial and should be addressed before this manuscript can be considered for publication at *eLife*.

1) While in principle, FIB-SEM is able to provide extremely well-aligned high-resolution data of neuropil, the data presented in the figures of this manuscript seemed to be of lower resolution and/or staining contrast than one would expect from nominally 8nm resolution imaging. Namely the few EM images in Figure 3 a) to d) have surprisingly little intracellular detail. For example, the mitochondria's internal structure is not visible. It may of course be that these are processed images that appear to be at lower resolution but this gives me a certain amount of concern about the ability to detect synapses in this data.

2) Synapse detection: The authors write a section on this topic, but it describes primarily how difficult synapse detection was -without clearly convincing that authors were able to detect synapses eventually (especially when the T-bar was missing). I did not understand how the authors calibrated the synapse detection in this data. Were high-resolution small image volumes obtained for direct comparison? Or were synaptic profiles of certain previously studied neurons compared, for instance to the Rybak or Tobin study?

3) Especially in the context of these concerns about the raw data it is in my view absolute standard by now that the raw image data and the reconstruction should be made freely available and browsable to reviewers and readers. As far as I can see the authors did not specify the dataset availability beyond the paper figures- also the transparent reporting sheet is actually largely empty. I am a bit surprised by this since obviously the Janelia team has a lot of resources to make EM data available to the community and this constitutes in my view an important aspect of such a publication. Especially also in light of the following points, I think it is even mandatory that the 3D image data and reconstructions are made available already at the review stage.

4) A few important quantifications are missing which are very relevant for putting this study in comparison to other connectomic studies:

a) What was the total annotation time invested to obtain these reconstructions (including curation and proof-reading time)?b) What is the total neuronal wiring length that is contained in the provided reconstruction?c) Density of reconstruction: The authors report a few qualitative statements ("no cell could hide"). What would be very helpful here is the volume fraction of neuropil that was accounted for by the ">250" (please give precise number) or 192 neurons. How does the number 93% , in the final paragraph of subsection “The synaptic dataset of glomerulus VA1v”, relate to the point about dense reconstruction discussed in the following paragraph?d) I find the description of synapse detection, second paragraph of subsection “Synapses” following, rather confusing.e) Is the number 87% , in subsection “The synaptic matrix of VA1v”, the volume fraction a accounted-for neuropil? This should be ideally reported in the section called "dense reconstruction" above.f) Numbers on error rates in the obtained Reconstructions are missing. How often did proof reading have to happen etc.?g) The section on reconstruction methods, is rather short and not very informative quantitatively.

5) I do get the impression that the manuscript does not report a large number of clear novel biological results. This is surprising since the authors state that connectivity of local neurons (LN) was not studied before. Even the Abstract, if I understand correctly, does not contain a biological finding beyond the reported reconstruction. I would therefore suggest considering this manuscript as a resource publication rather than a results manuscript. Alternatively, example analyses that would show the impact of this connectivity data would – at least for me as an outsider to this particular field – be very helpful in order to appreciate the depth and extent of this data.

As a final minor comment, I am surprised that the authors are citing the first but afterwards revised reconstruction of the fly medulla column, Takamura et al., 2013, instead of Takamura et al., 2017. In my understanding it is the 2017 *eLife* paper that provided the correct connectivity in that piece of fly neuropil.

Reviewer #3:

Horne and colleagues apply recent advances in focused ion beam milling scanning electron microscopy (FIB-SEM) to re-construct a largely complete connectome for the VA1v glomerulus in *Drosophila*. This is sexually dimorphic glomerulus is responsible for detecting female fly odors and is involved in signaling involving sex pheromones. In the last 3 years, considerable work has focused on reconstructing complete connectomes of olfactory regions by EM in the larval brain and indeed, in multiple olfactory glomeruli. However, these were accomplished largely using serial-section EM (ssEM); though ssEM can provide vast information, it can be difficult to clearly delineate small calibre neurites, thus confounding efforts at complete reconstruction. In using FIB-SEM to solve this problem, Horne and colleagues offer a more comprehensive dense reconstruction of a glomerulus, assigning most synapses to a parent neuron, and conducting a more thorough assessment of projection neurons and local interneurons.

The authors are to be commended on an outstanding contribution to *Drosophila* neurobiology; it is archival work that will be of great use to olfactory biologists. However, the deeper question is whether this is a significant advance to merit *eLife* publication. The FIB-SEM method has been established by a similar group of previous authors, so the methodology is not new. And though they were done with ssEM, this isn't even the first glomerulus to be fully reconstructed by EM. The new method does provide more information and more completion, but this is really an incremental increase. Because of these reasons, the novelty of the connectome is limited. Further, I don't have any intellectual issues with the analyses or the reconstructions, but I think this analysis would be more suited to a specialty journal.

The chief advances of this connectome rest in a much more comprehensive map of projection neuron and local interneuron projections. These are useful, but the scope is very narrow. Further, in the presentation of each type of neuron, the manuscript is written in such a way that non-fly-aficionados will have a tough time determining what's important. What are the new concepts that can be gleaned from this reconstruction? What do we learn about LN projections? What about PN projections? These areas are minimized (if not completely omitted), making it difficult to interpret why the new connectome is of value to neuroscience at large. These issues would greatly enhance the readability of the manuscript, though I still feel the core advance represented by the paper is better suited for another journal.

[Editors' note: further revisions were requested prior to acceptance, as described below.]

Thank you for resubmitting your work entitled "A resource for the *Drosophila* antennal lobe provided by the connectome of glomerulus VA1v" for further consideration at *eLife*. Your revised article has been favorably evaluated by Eve Marder (Senior Editor), a Reviewing Editor, and 3 reviewers. The reviewers have discussed their opinions about the revised manuscript, and are in agreement.

The manuscript has been improved but there are some remaining issues that need to be addressed before acceptance, as outlined below.

Most importantly, the raw data needs to be available for inspection by reviewers before we can accept the paper. Please send a revised manuscript that addresses the rest of the issues (mostly minor) below, after the data are uploaded to a website that the reviewers can inspect. Thank you.

Reviewer #1:

The revised manuscript by Horne et al. is much improved in its presentation and accessibility to a general readership. Also the new format of "Tools and Resources" fits better with the dataset. So I am enthusiastic in support its publication, assuming that the EM expert reviewer is satisfied with the technical aspect of the data.

Below are a few (minor) suggestions for further improvement.

1) Since the authors emphasize in the Abstract that VA1v is a sexually dimorphic glomerulus, they should tell the readers more explicitly in the main text that the sample is from a female (right now the information is in the Materials and methods and figure legend), and what is the nature of sexual dimorphism (male VA1v has larger volume than female VA1v).

2) Subsection “Identified neurons from dense reconstruction in VA1v2”, paragraph three: most cells (should be changed to LNs) being panglomerular. On that note, none of the LNs in reconstructed LNs in Figure 4—figure supplement 3 appeared panglomerular, presumably because they were not fully reconstructed; the authors should state this explicitly, and how they categorize the LNs even though they were not fully reconstructed.

3) Figure 7A: it would be useful to add an output arrow from PNs, the destination being "higher olfactory centers." Even though it is not part of the reconstruction, it will be useful to remind the readers where the information goes from the antennal lobe.

Reviewer #2:

The revision of Horne et al. is improved, the formatting as a resource paper serves the data much better. The text has improved and contains more quantifications. Synapse detection is properly described and convincing.

As to the availability of the raw image data at the review stage, I maintain the view that by *eLife* standards (and similarly for many other journals) the raw data should be made available to reviewers for inspection, not just promised to be available later. This is especially true for a resource manuscript, but applies more broadly. I would hope that with the infrastructural capabilities of Janelia, data from the FlyEM team could be rather spearheading such modern data transparency standards than circumventing them.

As to the reporting of quantifications, I am surprised that the authors choose NOT to report certain quantitative measurements. The authors state that the reconstruction comprised a total of "15.8 cm of neurite". Since the authors are free to put this number into context, as in the reply to my comment, I would encourage this number to be reported, not withheld from the reader.

Reviewer #3:

The revised version of Horne et al. is a much-improved manuscript. Overall, I am satisfied with the comments from the authors and the considerable revisions included therein. I think the adjustment to a "Tools and Resource" and the reframing of many of the conceptual advances from this manuscript are absolutely appropriate. I thank the authors for their consideration of most of the comments provided.

I will defer judgment to many of the mathematical and technical issues raised by the second reviewer, however, as these fall out of my purview.

One confusing section, however, is the third paragraph of subsection “Identified neurons from dense reconstruction in VA1v”. It seems like the goal is to highlight aspects of ORNs, but some of it seems to be discussing LNs. The LN cells identified by Chou et al., are panglomerular, but the wording makes it sound like the ORNs are pan-glomerular. Is this accurate? Could the paragraph be adjusted to improve clarity?

---

## [Author Response]

Essential revisions:- Please submit as a Tools and Resource paper (given reviewers' reservations about new biological insight)

We have revised our article to conform to the Instructions for a ‘Tools and Resource’ paper and, reflecting this change in emphasis, now revise the title to:

A resource for the *Drosophila* antennal lobe provided by the connectome of glomerulus VA1v

We take notice that Tools and Resources articles “do not have to report major new biological insights or mechanisms, but it must be clear that they will enable such advances to take place. Specifically, submissions will be assessed in terms of their potential to facilitate experiments that address problems that to date have been challenging or even intractable.” We therefore address the widely evoked role for connectomic data in enabling functional studies on identified neuron circuits, in the following original text:

“Including the synaptic network of all LNs and PNs to generate an actual map of the complete synaptic network, or connectome (Lichtman and Sanes, 2008), for each glomerulus is thus a final step towards adopting functional connectomic approaches (Venken et al., 2011; Meinertzhagen and Lee, 2012) to the analysis of antennal lobe function.” see also the following text, now added in the Discussion: “The connectome we report is essential to promote functional analyses using genetic dissection methods (Venken et al., 2011). In particular it will enable the interpretation of future analyses of network function made possible […]”

We summarize our findings with the general short statement: “Unlike PNs and their ORN inputs, and except LN2V, we find that LNs have three main characteristics: they have fewer synapses on average than PNs and ORNs; these are heterogeneous, and they form undirected, reciprocal synaptic networks which are inhibitory.”

“The connectome we report is essential to promote functional analyses using genetic dissection methods (Venken et al., 2011). In particular it will enable the interpretation of future analyses of network function made possible: first, by imaging methods, either in vitro, especially using genetically encoded calcium (e.g. Tian et al., 2012) or voltage (e.g. Antic et al., 2016) indicators; or second, during intact behaviour in vivo (e.g. Grover et al., 2016), in response to single odours or odour combinations (e.g. Silbering and Galizia, 2007). Our data also support computational approaches to insect olfaction.”

We have already emphasized that our EM connectomic data reveal that previous genetic screens of antennal lobe cells were clearly not saturated. This is an important finding for evaluating past reports in the field, which have relied heavily on genetic screens to reveal the antennal lobe’s component cells, but which our report now shows to be incomplete, and thus not wholly reliable. We do not wish to attack previous reports, which have contributed much, but in three places we refer quite clearly to the lack of saturation in previous screens (see Abstract, Discussion, or search “saturat” in the text), and regard this as a major scientific finding of our study.

- Please provide synaptic calibration data (report precision/recall numbers on what they call a synapse) and other relevant metrics

We now report these metrices in several places, as follows:

A newly combined section headed Synapse annotationin the Materials and methods where we report: “[…] trained proofreaders who had attained a recall proficiency in excess of 85% and a precision of >94.7% were used to annotate presynaptic T-bars in the EM volume.” Further information continues on in this paragraph, some parts of which were moved from the Results.

- Please ensure full data availability, which will be checked by reviewers/editors before final acceptance

We have now included the complete matrix of all single-cell synaptic partnerships as a data file, Figure 5—figure supplement 1. This is the burden of our complete dataset, suitable for computational studies, as well as the basis to interpret future functional analyses using methods based on genetic reporters.

- Please re-write the manuscript with the general readership of eLife in mind, integrating as much as possible previous data on the subject and providing biological significance of the finding.

We have endeavoured to do this by highlighting more clearly what we see as the novel, especially functional value of our anatomical analysis, but also take notice that “Tools and Resources articles do not have to report major new biological insights or mechanisms”. We do also endeavour to make it “clear that they will enable such advances to take place. Specifically, submissions will be assessed in terms of their potential to facilitate experiments that address problems that to date have been challenging or even intractable.” Our data fall, we feel, into this category, as we now indicate in several places in the text, as follows:

1) “The connectome we report is essential to promote functional analyses using genetic dissection methods (Venken et al., 2011). In particular it will enable the interpretation of future analyses of network function made possible: first, by imaging methods, either in vitro, especially using for example genetically encoded calcium (Tian et al., 2012) or voltage (Antic et al., 2016) indicators, or second, during intact behaviour in vivo (e.g. Grover et al., 2016), in response to odours or odour combinations (e.g. Silbering and Galizia, 2007). Our data also support computational approaches to insect olfaction.” (Discussion, paragraph 2)

2) “A prospect made possible by the connectome we report here”

3) Finally, we indicate that the genetic screens so widely used to date, seem to underestimate PN numbers considerably, and are thus not saturated, leading us to conclude that “This is a major conclusion to be drawn from our connectome.”

- Please also address other comments from the reviewers; see below for the full reviews.

Please see our responses to each comment in the rather long list given below. We indicate where we find that the reviewers’ suggestions are best met by the altered emphasis in our submission provided by its revision to a Tools and Resource paper, for which the original text was, in any case, already better suited.

We also shortened the text of the Discussion by removing a rather inconclusive terminal paragraph comparing the vertebrate olfactory bulb and fly antennal lobe, and cut:

“Limited comparisons with the circuits of the mammalian olfactory bulb are possible (Hildebrand and Shepherd, 1997). Two features are immediately obvious, the diversity of cell types and the complexity of their circuits. PN diversity in *Drosophila* is reminiscent of the projection neurons cohabiting individual glomeruli in the olfactory bulb. These are of two types, mitral cells and tufted cells, each with different morphological types and each projecting to different regions in the higher brain (Nagayama et al., 2014), much as PN cells do. The granule cell interneurons of the olfactory bulb share circuit similarities with *Drosophila* LN cells in being connected reciprocally to mitral and tufted output neurons, inhibiting these and receiving excitation from them (Shepherd and Greer, 1998). In *Drosophila* many LNs are correspondingly GABAergic (Wilson and Laurent, 2005; Okada et al., 2009) and thus likely to mediate inhibition.”

Reviewer #1:[…] 1) My major critique is that the current writing style makes it very difficult for readers besides a small group of specialists to be interested in reading the paper. The description of the data is accompanied with little biological meanings or insights. While EM reconstruction must describe dry data, this one is an extreme; for comparison, Tobin et al., 2016 is much more interesting to read even for a specialist. Major efforts need to be put to streamline the writing and interpret the data with biological context, with the general readership of eLife in mind.

The senior author takes responsibility for the previous version of the text. We have endeavoured to address all points and related comments from the other reviewers by revising much of the text so as to rearrange, improve and promote the flow of ideas, as indicated in red in the attached manuscript, and in particular as follows: 1) we have compiled all numerical summaries of our findings, which hitherto were scattered under different headings, into a single section in the Results. We present this summary in the section “The synaptic matrix of VA1v” in which we report the complete connectome, and make only passing reference to it in two other places in the Results, citing the connectome as a cross-reference.

1) We further emphasize the completeness of our synaptic analysis, especially of the numerous LNs, which have not previously been documented at synaptic level.

2) Given that we now submit our study as a Tools and Resource paper, we now expand the synaptic database originally presented as the entire matrix in Figure 5 by adding an additional figure supplement as a new look-up spreadsheet (Figure 5—figure supplement 1), which provides the numerical intercepts of the matrix in readable form. See previous response to this reviewer.

2) There are a few significant errors or vague statements that further impede reading. The most important one occurs in paragraph three of subsection “Identified neurons and their synaptic networks”. I believe "medial lateral tract" should be "medial tract", or mALT. The medial lateral tract contains entirely different type of neurons (belonging to group b). Furthermore, in the same paragraph, it promises three groups, but I did not see c) after a) and b).

We have amended medial lateral tract to medial tract, mALT, and reassigned the cell groups to become a)-c) with subtypes i)-iii). Our apologies for careless writing.

3) Assuming my interpretation above is correct, there are only 3 uniglomerular PNs that belong to the mALT group. This number is different from the reported 5 based on lineage tracing using dual-color MARCM (Yu et al., PLoS Biology 2010). The authors should comment on this. In general, it will be useful for the authors to systematically specify which neurons in their reconstruction have been traced to the cell bodies, and what is the location of cell bodies with respect to the antennal lobe neuropil; such information can be presented an additional column or row in Figure 5. These will provide cross-validation as the lineages of antennal lobe neurons have been extensively documented in the literature.

We were mortified to see our omission of the report by Yu et al. (2010), which we now include in our revision.

On the other hand, we believe that the reason for the difference in the number of PNs (we report 3 in VA1v and they report 5) is because the border between VA1v and the neighbouring glomerulus is not correct in the report by Yu et al. We therefore believe that the 5 reported by these authors are in fact the 3 that we find do enter VA1v plus 2 that enter a neighbouring glomerulus. In short, we think the discrepancy is a border dispute between close-neighbouring glomeruli. Moreover we cannot imagine how two entire PNs from our dataset could have been overlooked: there are nowhere sufficient orphan profiles to constitute two complete large neurons. One possible reconciliation might of course be if different flies have glomeruli of different cellular compositions, but this would not have invalidated conclusions from our dataset. We therefore think the correct attribution of borders between neighbouring glomeruli is the problem, and now state this explicitly in the text.

We have now specified those neurons that have been traced to their somata, and where possible indicated the locations of the latter. Cell bodies are now included as a column in Figure 2-source data 1. We were unable to trace a number of cells back to the cell body of origin, if this lay outside the grey scale ROI. This was in any case a lot of work.

4) Speaking of Figure 5, I need to amplify to about 600% on my large computer screen in order to see the data. I expect that it will be quite difficult to fit into a regular eLife page with sufficient resolution. Figure 8 suffers from the same problem. The authors should think of creative ways of presenting these data that are more readily seen by readers.

Yes, we agree with the reviewer concerning this problem, which we share with other contemporary accounts not least with one by Chou et al., 2010, for which the senior author is Liquin, who present a large matrix, the same size as our Figure 5, which similarly can be read only at 600%, but even then we think not as clearly as ours! The problem is numerical and not of our making, but the reviewer seeks resolution notwithstanding. We considered and reconsidered this problem, trying for example to address the issue by collapsing individual cells, in columns and rows, into cell types, as opposed to single cells. In fact this is not very satisfactory either, because it assigns equal space to columns that contain a great many cells (ORNs especially) as to columns that report only a single cell. To provide the complete matrix for single cells, we therefore wish still to give the complete data for these as a single-cell matrix, as well as in a more readable spreadsheet in supplementary data (see Figure 5—figure supplement 1), enabling interested readers to download our complete dataset for further analyses offline (see comment above). This is the most satisfactory solution we could reach, and can think of no alternative, nor apparently could the reviewer.

5) The manuscript contains a number of citation errors. Two examples: Introduction paragraph four, they should cite Marin et al., 2002, which was published back-to-back with Wong et al., and which contains several unique single cell tracings that are relevant to this study. In paragraph seven of subsection “Identified neurons and their synaptic networks”, I think they mean Tanaka 2012 (rather than 2010).

We now also cite Marin et al., 2002, with apologies for having omitted it, and have amended Tanaka et al., 2010 to 2012. We also additionally cite Silbering and Galizia, 2007, citation of whose work on LNs was previously omitted

6) The acronym FIB-SIM should be spelled out in Abstract at the first mention. The authors should also consider adding "nearly" before "complete" to their title given that there are still orphan profiles (despite the tremendous amount of work!).

We prefer not to spell out FIB-SEM in the Abstract, which would have increased the word total beyond the permissible limit, and reference to FIB-SEM then appears in the Results, where it is indeed spelled out. For the title, and partly for stylistic reasons, we really prefer not to add ‘nearly’ but rather to remove ‘complete’ so as to eliminate all reference to completeness. There are in fact several reasons why our connectome, like all others, is not complete, chiefly in having elements we were unable to identify, but the level of our completeness compares very favourably with contemporary reports in other connectomes. Given that we have redirected our submission to a Tools and Resource paper, we indicate the new emphasis anyway by a change in the title: A resource for the *Drosophila* antennal lobe provided by the connectome of glomerulus VA1v

Reviewer #2:The manuscript "The complete connectome of a Drosophila antennal lobe glomerulus" by Horne et al. reports the electron-microscopic imaging and dense connectomic reconstruction of the synaptic circuits in a glomerulus of the antennal lobe in Drosophila melanogaster.This is a comprehensive piece of work containing a lot of potentially useful connectomic information.In my understanding, compared to previous connectomic reconstructions of glomeruli in Drosophila which the authors cite (Rybak et al., Tobin et al.), this study includes the connectivity of local interneurons which were not available in the previous publications.

Yes, this is by far the most notable feature of our study.

Since I am neither an expert in the olfactory system nor Drosophila neurobiology I will focus my comments on the EM and connectomic reconstruction aspects of this work.This is a comprehensive piece of data and I commend the authors for this work, but I have a few concerns that in my view are substantial and should be addressed before this manuscript can be considered for publication at eLife.1) While in principle, FIB-SEM is able to provide extremely well-aligned high-resolution data of neuropil, the data presented in the figures of this manuscript seemed to be of lower resolution and/or staining contrast than one would expect from nominally 8nm resolution imaging. Namely the few EM images in Figure 3 a) to d) have surprisingly little intracellular detail. For example, the mitochondria's internal structure is not visible. It may of course be that these are processed images that appear to be at lower resolution but this gives me a certain amount of concern about the ability to detect synapses in this data.

There are several things in this comment to which we should respond. As the reviewer indicates, the resolution is not that of TEM, but FIB-SEM. It is true that sampling at 8nm per pixel should be sufficient to reveal the cristae of mitochondria in appropriately aligned images. In fact it is possible to see striations in the mitochondrion at 9 o’clock in Figure 3A, although we agree these are not clear, certainly not as they would be in a single TEM image.

There are two major issues. 1) First is the imaging method. In our images the grey level of the synapse has been saturated so as to see membrane details more clearly, but in reality the synapses are ~2x darker than the membranes, which makes their electron density easier to distinguish as well. For that reason we sacrificed resolution to gain membrane contrast, and increase ease of later segmentation and proofreading steps, and to this extent the images are processed as the reviewer suggests. 2) In addition, the fixation method we have used, light aldehyde fixation followed by high-pressure freezing was in fact selected empirically over many trials to highlight membranes and synapses clearly. We now modify the text to indicate the fact that synapses stand out very clearly by the density of their staining, so that we do not need to rely so much on resolving the substructural details of the synaptic organelles as in TEM images of *Drosophila* or possibly in mammalian tissue. We hope the reviewer will agree that synapses in Figure 3 are simply unmistakable as dark bodies, and that what is missing in most cases is the clear platform that characterizes fly synapses, but not those of other insect species.

2) Synapse detection: The authors write a section on this topic, but it describes primarily how difficult synapse detection was -without clearly convincing that authors were able to detect synapses eventually (especially when the T-bar was missing). I did not understand how the authors calibrated the synapse detection in this data. Were high-resolution small image volumes obtained for direct comparison? Or were synaptic profiles of certain previously studied neurons compared, for instance to the Rybak or Tobin study?

What is important about our methods is that the synapses (presynaptic T-bars) are electron dense and clearly visible as dark structures, and even though they may lack a clear platform they have a very distinct pedestal. The Rybak and Tobin studies both used TEM images in which the platform is clear, but are otherwise similar to those we see with FIB. However, both studies analysed different neurons than those we have identified. Rybak reported circuits from three different glomeruli (DM2, DL5, and VA7) while Tobin reported DM6 from several brains, but neither previous study reported cells from vA1V.

Why no platform on the T-bar? We explain this further in the following sentence as follows: “This chiefly resulted because in our FIB-SEM images the grey level of the synapses was about twice that of the membranes. We therefore adjusted the electron density of images to increase membrane contrast, because this proved advantageous to enhance membrane continuity more reliably during later proof-reading steps (see Materials and methods), but rendered the platform of the T-bar ribbon often less distinctly in our FIB-SEM images than when seen in TEM.”

Are we accurate? We calibrated synapse detection in a number of ways:

1) First, the annotators were specifically trained to detect these profiles, and their numbers are comparable to those seen by Tobin, as the number of presynaptic T-bars per ORN. We cite the comparison:

“[…] of ipsilateral ORNs to mPN1 21.6 ± 7.7 and for contralateral ORNs to mPN1 is 9.2 ± 4.6 synapses, compared with a recent report (Tobin et al., 2017) which found an average of 23 synapses from ipsilateral ORNs” We could not make more exact comparisons with these two previous studies, because we examined a different glomerulus and cells than did either of these two.

2) Exact comparisons are hard to make because the synaptic data are differently reported, but we can compare our density of synaptic sites with that reported by Rybak using ssEM and we find similar numbers overall. Thus we found “roughly 2.29 synapses per µm^3^” which “corresponds approximately to the differently computed values reported from TEM by Rybak et al., 2016 for glomerulus VA7”.

3) Especially in the context of these concerns about the raw data it is in my view absolute standard by now that the raw image data and the reconstruction should be made freely available and browsable to reviewers and readers. As far as I can see the authors did not specify the dataset availability beyond the paper figures- also the transparent reporting sheet is actually largely empty. I am a bit surprised by this since obviously the Janelia team has a lot of resources to make EM data available to the community and this constitutes in my view an important aspect of such a publication. Especially also in light of the following points, I think it is even mandatory that the 3D image data and reconstructions are made available already at the review stage.

We will of course provide raw data and 3D image data, from the Janelia link (emdata.janelia.org/VA1v) to the greyscale and segmentation data. We have also added the following files into the transparent reporting sheet, as requested by the reviewer: Figure 5—figure supplement 1, and an Excel spreadsheet for Figure 2-source data 1, replacing Table 2.

4) A few important quantifications are missing which are very relevant for putting this study in comparison to other connectomic studies:a) What was the total annotation time invested to obtain these reconstructions (including curation and proof-reading time)?

We have now added these times: “Our analysis took 60 person months of proofreading time and 20 person months for curation.” In the Materials and methods under the section on Reconstruction.

b) What is the total neuronal wiring length that is contained in the provided reconstruction?

The 192 bodies densely reconstructed in the glomerulus and sparsely traced outside it comprise 15.8 cm of neurite. We think this measure is a poor representation of neuron complexity and reconstruction effort for our connectome, however, given 1) the extreme size differences between different species; 2) that most of the complexity of the reconstruction resides in small processes by synapses; and 3) that the goal is finding connections not achieving wirelength. The total number of synapses reconstructed was >11,140 presynaptic sites with ~38,050 postsynaptic dendrites, as reported. Given our view, we elect not to include wirelength as a relevant measure but provide it to the reviewer for reference.

c) Density of reconstruction: The authors report a few qualitative statements ("no cell could hide"). What would be very helpful here is the volume fraction of neuropil that was accounted for by the ">250" (please give precise number) or 192 neurons. How does the number 93% , in the final paragraph of subsection “The synaptic dataset of glomerulus VA1v”, relate to the point about dense reconstruction discussed in the following paragraph?

We do report that 87% of the volume of glomerulus VA1v was listed in our connectome of 192 neurons. We also indicate that 93% of connections were resolved with both pre- and postsynaptic partners identified, including some connecting in neighbouring glomeruli. 87.9% of the synaptic connections have been assigned to the neurons in the connectome, leaving 5% of synapses connecting neurons not included in the connectome because they had <50 synapses.

d) I find the description of synapse detection, second paragraph of subsection “Synapses” following, rather confusing.

We have addressed this important point in our responses to comment 1 from this same reviewer. See para beginning “Why no platform on the T-bar?”

e) Is the number 87% , in subsection “The synaptic matrix of VA1v”, the volume fraction a accounted-for neuropil? This should be ideally reported in the section called "dense reconstruction" above.

We removed text under the heading dense reconstruction and coupled it with other metrical data under the section: The synaptic matrix of VA1v.

f) Numbers on error rates in the obtained Reconstructions are missing. How often did proof reading have to happen etc.?

Proof reading was undertaken continuously by trained proof-readers, who undertook dense reconstruction of an entire volume of neuropile, not redundant sparse tracing of that neuropile. We compared most of our LN reconstructions with those of the Wilson lab (Harvard) done by Asa Barth-Maron using sparse reconstruction of the same glomerulus and image series.

g) The section on reconstruction methods, is rather short and not very informative quantitatively.

These methods are based on those used elsewhere at Janelia and are well documented and cited in other reports (e.g. Takemura et al., 2017). We provide additional detail in this section and six references.

5) I do get the impression that the manuscript does not report a large number of clear novel biological results. This is surprising since the authors state that connectivity of local neurons (LN) was not studied before. Even the Abstract, if I understand correctly, does not contain a biological finding beyond the reported reconstruction. I would therefore suggest considering this manuscript as a resource publication rather than a results manuscript. Alternatively, example analyses that would show the impact of this connectivity data would – at least for me as an outsider to this particular field – be very helpful in order to appreciate the depth and extent of this data.

We now provide additional biological conclusions in the Abstract in which we hypothesize the mutual reciprocal inhibition by LNs onto PNs, and also between LNs in different glomeruli. Connections are sparse and variation in the strengths of connections. LNs have four distinguishing characteristics: unlike PNs and their ORN inputs, except LN2V, LNs have three main characteristics: they have fewer synapses on average than PNs and ORNs, have heterogeneous synaptic connections, and are undirected and reciprocal in their synaptic networks. Given the suggestion of the reviewing editor, we have now indeed revised our resubmission as a Tools and Resource paper.

As a final minor comment, I am surprised that the authors are citing the first but afterwards revised reconstruction of the fly medulla column, Takamura et al., 2013 Nature, instead of Takamura et al., 2017. In my understanding it is the 2017 eLife paper that provided the correct connectivity in that piece of fly neuropil.

We appreciate this point but should make it clear that the 2013 report was not in fact incorrect, merely incomplete, because it contained only a single medulla column and because some inputs arise in neighbouring columns that as a result could not be traced. The correction is not merely semantic, because the 2013 paper was a landmark advance that for the first time identified pathways from lamina inputs in the distal medulla to outputs onto T4 in the proximal medulla, laying the groundwork for solving the biological implementation of the H-R EMD. Looking through the text, there are many references to the 2015 paper, but in any case we did not find a citation to the 2013 paper that we think is misplaced and should be changed, so we are otherwise at a loss to know how to respond.

Reviewer #3:Horne and colleagues apply recent advances in focused ion beam milling scanning electron microscopy (FIB-SEM) to re-construct a largely complete connectome for the VA1v glomerulus in Drosophila. This is sexually dimorphic glomerulus is responsible for detecting female fly odors and is involved in signaling involving sex pheromones. In the last 3 years, considerable work has focused on reconstructing complete connectomes of olfactory regions by EM in the larval brain and indeed, in multiple olfactory glomeruli. However, these were accomplished largely using serial-section EM (ssEM); though ssEM can provide vast information, it can be difficult to clearly delineate small calibre neurites, thus confounding efforts at complete reconstruction. In using FIB-SEM to solve this problem, Horne and colleagues offer a more comprehensive dense reconstruction of a glomerulus, assigning most synapses to a parent neuron, and conducting a more thorough assessment of projection neurons and local interneurons.

We do not agree with the reviewer’s viewpoint. There has been no prior comprehensive dense reconstruction of a glomerulus, only of the ORN and PN circuits. The more numerous local neurons of the glomerulus have never been comprehensively reconstructed, nor their circuits identified. This was clearly stated in the first paragraph of the Discussion, now retained in the revised text.

The authors are to be commended on an outstanding contribution to Drosophila neurobiology; it is archival work that will be of great use to olfactory biologists. However, the deeper question is whether this is a significant advance to merit eLife publication. The FIB-SEM method has been established by a similar group of previous authors, so the methodology is not new. And though they were done with ssEM, this isn't even the first glomerulus to be fully reconstructed by EM.

Again, we do not quite agree on this point. This is the first glomerulus to be fully reconstructed by EM. Previous reports made important contributions by covering specific PN and ORN contributions to a glomerulus, or three glomeruli, but the numerous, widespread synaptic circuits of LNs have not previously been systematically reported. As we state in the first sentence of the Discussion “Our study reports the complete synaptic connectome for all three cell types, ORN, LN and PN, […] providing a proof of principle for the remaining 50 glomeruli. It complements two contemporary EM studies … (Rybak et al., 2016; Tobin et al., 2017), augmenting these by including important synaptic circuits of the LNs, […] which constitute nearly 30% of the cells of the glomerulus, and much of its synaptic diversity.”

The new method does provide more information and more completion, but this is really an incremental increase.

We of course are the authors, but we do feel this comment of the reviewer is as incorrect as it is damaging. Together with the PNs and ORNs for glomerulus vA1v we report the connections of 56 local neurons. Previous reports from TEM, while themselves important additions, have been able to document very few if any connections of the LNs, which we provide based on improved FIB-SEM methods, and therefore claim that the increment to which the reviewer refers is in fact very large (56 additional entire cells, all of them inaccessible to TEM methods, and most of them synaptically diverse).

Because of these reasons, the novelty of the connectome is limited.

This may well be the reviewer’s perspective, but we suggest it is not based on the size of the connectome we have now generated.

Further, I don't have any intellectual issues with the analyses or the reconstructions, but I think this analysis would be more suited to a specialty journal.

This may likewise be true, but our report is, we suggest, at least equal in importance to the excellent recent *eLife* report by Tobin et al.

The chief advances of this connectome rest in a much more comprehensive map of projection neuron and local interneuron projections. These are useful, but the scope is very narrow. Further, in the presentation of each type of neuron, the manuscript is written in such a way that non-fly-aficionados will have a tough time determining what's important. What are the new concepts that can be gleaned from this reconstruction?

We believe that the comprehensive quantitative analysis of the extent of reciprocity between partner neurons is both novel and important, and has not been previously reported comprehensively for any other neuropile.

What do we learn about LN projections? What about PN projections? These areas are minimized (if not completely omitted), making it difficult to interpret why the new connectome is of value to neuroscience at large. These issues would greatly enhance the readability of the manuscript, though I still feel the core advance represented by the paper is better suited for another journal.

We learn that the LN projections are primarily reciprocal and formed with other LNs, which was not previously known, and we now emphasize this conclusion more strongly.

[Editors' note: further revisions were requested prior to acceptance, as described below.]

The manuscript has been improved but there are some remaining issues that need to be addressed before acceptance, as outlined below.Most importantly, the raw data needs to be available for inspection by reviewers before we can accept the paper. Please send a revised manuscript that addresses the rest of the issues (mostly minor) below, after the data are uploaded to a website that the reviewers can inspect. Thank you.

We agree unreservedly and Dr Steve Plaza, a co-author, has now provided the following information:

“FlyEM has and will continue to provide unprecedented and timely access to our datasets as evidenced by the public datasets on emdata.janelia.org. In this case, we needed some additional time as we are refactoring our infrastructure to provide ability to access the data in the most optimal ways. There are (only a) few good Big Data tools to access connectomic data, which is why FlyEM spends resources developing such cutting edge technology. While most publications believe a directory of images and dump of a large spreadsheet is sufficient, we feel this to be a grotesque way to release such large and beautiful data. We have not finished this refactoring of our infrastructure but have provided a relatively functional but not fully featured release at http://emdata.janelia.org/AL-VA1v. For more information go to the Readme file. For those desiring to see the data through a web viewer please visit:

http://neuroglancer-demo.appspot.com/#!%7B%22layers%22:%7B%22grayscalejpeg%22:%7B%22source%22:%22dvid://http://35.199.29.12:8600/ab6e610d4fe140aba0e030645a1d7229/grayscalejpeg%22%2C%22type%22:%22image%22%7D%2C%22segmentation%22:%7B%22source%22:%22dvid://http://35.199.29.12:8000/d925633ed0974da78e2bb5cf38d01f4d/segmentation%22%2C%22type%22:%22segmentation%22%7D%7D%2C%22navigation%22:%7B%22pose%22:%7B%22position%22:%7B%22voxelSize%22:%5B8%2C8%2C8%5D%2C%22voxelCoordinates%22:%5B3164.243896484375%2C7989.96728515625%2C3391%5D%7D%7D%2C%22zoomFactor%22:76.76400940694718%7D%2C%22perspectiveOrientation%22:%5B0.1018502488732338%2C-0.1967611163854599%2C0.00035121332621201873%2C0.9751468896865845%5D%2C%22perspectiveZoom%22:121.51041751873501%2C%22layout%22:%224panel%22%7D

and

http://neuroglancer-demo.appspot.com/#!%7B%22layers%22:%7B%22grayscalejpeg%22:%7B%22source%22:%22dvid://http://35.199.29.12:8600/ab6e610d4fe140aba0e030645a1d7229/grayscalejpeg%22%2C%22type%22:%22image%22%7D%2C%22segmentation%22:%7B%22source%22:%22dvid://http://35.199.29.12/d925633ed0974da78e2bb5cf38d01f4d/segmentation%22%2C%22type%22:%22segmentation%22%2C%22segments%22:%5B%22205788%22%2C%223%22%2C%22729439%22%5D%7D%7D%2C%22navigation%22:%7B%22pose%22:%7B%22position%22:%7B%22voxelSize%22:%5B8%2C8%2C8%5D%2C%22voxelCoordinates%22:%5B3752.35009765625%2C8052.685546875%2C3647.19970703125%5D%7D%2C%22orientation%22:%5B-0.02181488275527954%2C0%2C0%2C0.9997619986534119%5D%7D%2C%22zoomFactor%22:29.520611734760028%7D%2C%22perspectiveOrientation%22:%5B0.13834506273269653%2C0.5922386050224304%2C0.7721632719039917%2C0.18405957520008087%5D%2C%22perspectiveZoom%22:864.801499283442%2C%22layout%22:%224panel%22%7D

The first link shows a view similar to Figure 2, and the second reconstructed ORN, mPN1 and LN2L cells.

To access the data programmatically (more documentation forthcoming) consult documentation for github.com/janelia-flyem/dvid. In the future, we expect much better strategies for accessing the data and distributing it through cloud services, which will facilitate better scientific reproducibility, something that has been generally absent in most connectomic works to date.

We corrected an editing error in the legend to Figure 5.

We now also cite a recent report that has appeared since our initial submission, on the developmental origins of LNs, adding the following text: “possibly reflecting the three developmental modes of their origin, as residual larval LNs, as adult-specific LNs emerging before associated sensory and projection neurons, and as LNs that emerge after synaptic connections are established (Liou et al., 2018).”

In addition, we have added a new reference and citation to support this revision:

Liou et al., 2018.

Reviewer #1:[…] 1) Since the authors emphasize in the Abstract that VA1v is a sexually dimorphic glomerulus, they should tell the readers more explicitly in the main text that the sample is from a female (right now the information is in the Materials and methods and figure legend), and what is the nature of sexual dimorphism (male VA1v has larger volume than female VA1v).

We indicated at the start of the Results section that we used a female *Drosophila*, and moved a short paragraph from the Materials and methods to the first paragraph of the Results, as follows: “a *fruitless*-positive glomerulus responsive to fly odour (Sakurai et al., 2013), that signals the sex pheromones cis-vaccenyl acetate and methyl laurate (Kurtovic et al., 2007; Dweck et al., 2013, and is significantly larger in male flies than in females (Kondoh et al., 2003; Stockinger et al., 2005).”

2) Subsection “Identified neurons from dense reconstruction in VA1v2”, paragraph three: most cells (should be changed to LNs) being panglomerular. On that note, none of the LNs in reconstructed LNs in Figure 4—figure supplement 3 appeared panglomerular, presumably because they were not fully reconstructed; the authors should state this explicitly, and how they categorize the LNs even though they were not fully reconstructed.

We now explicitly state that our reconstructions of LNs report only those portions in VA1v and additional regions sufficient to identify the soma and its axon, and as a result are therefore still partial.

3) Figure 7A: it would be useful to add an output arrow from PNs, the destination being "higher olfactory centers." Even though it is not part of the reconstruction, it will be useful to remind the readers where the information goes from the antennal lobe.

We have added an arrow to the figure and indicated its significance in the legend to this figure.

Reviewer #2:The revision of Horne et al. is improved, the formatting as a resource paper serves the data much better. The text has improved and contains more quantifications. Synapse detection is properly described and convincing.As to the availability of the raw image data at the review stage, I maintain the view that by eLife standards (and similarly for many other journals) the raw data should be made available to reviewers for inspection, not just promised to be available later. This is especially true for a resource manuscript, but applies more broadly. I would hope that with the infrastructural capabilities of Janelia, data from the FlyEM team could be rather spearheading such modern data transparency standards than circumventing them.

See our response above.

As to the reporting of quantifications, I am surprised that the authors choose NOT to report certain quantitative measurements. The authors state that the reconstruction comprised a total of "15.8 cm of neurite". Since the authors are free to put this number into context, as in the reply to my comment, I would encourage this number to be reported, not withheld from the reader.

We have now added this metric in subsection “Identified neurons from dense reconstruction in VA1v”.

Reviewer #3:The revised version of Horne et al. is a much-improved manuscript. Overall, I am satisfied with the comments from the authors and the considerable revisions included therein. I think the adjustment to a "Tools and Resource" and the reframing of many of the conceptual advances from this manuscript are absolutely appropriate. I thank the authors for their consideration of most of the comments provided.I will defer judgment to many of the mathematical and technical issues raised by the second reviewer, however, as these fall out of my purview.One confusing section, however, is the third paragraph of subsection “Identified neurons from dense reconstruction in VA1v”. It seems like the goal is to highlight aspects of ORNs, but some of it seems to be discussing LNs. The LN cells identified by Chou et al., are panglomerular, but the wording makes it sound like the ORNs are pan-glomerular. Is this accurate? Could the paragraph be adjusted to improve clarity?

We agree: this was a word-processing error and the text now reads: A total of 51 ORNs originated in the ipsilateral antennal nerve with 56 that entered in the commissure from the contralateral lobe.